# Proximity labeling of protein complexes and cell-type-specific organellar proteomes in *Arabidopsis* enabled by TurboID

**Andrea Mair[1,2], Shou-Ling Xu[3], Tess C Branon[1,4,5,6], Alice Y Ting[1,5,6,7], Dominique C Bergmann[1,2]\***

[1]Department of Biology, Stanford University, Stanford, United States; [2]Howard Hughes Medical Institute, Chevy Chase, United States; [3]Department of Plant Biology, Carnegie Institution for Science, Stanford, United States; [4]Department of Chemistry, Massachusetts Institute of Technology, Cambridge, United States; [5]Department of Genetics, Stanford University, Stanford, United States; [6]Department of Chemistry, Stanford University, Stanford, United States; [7]Chan Zuckerberg Biohub, San Francisco, United States

**Abstract** Defining specific protein interactions and spatially or temporally restricted local proteomes improves our understanding of all cellular processes, but obtaining such data is challenging, especially for rare proteins, cell types, or events. Proximity labeling enables discovery of protein neighborhoods defining functional complexes and/or organellar protein compositions. Recent technological improvements, namely two highly active biotin ligase variants (TurboID and miniTurbo), allowed us to address two challenging questions in plants: (1) what are in vivo partners of a low abundant key developmental transcription factor and (2) what is the nuclear proteome of a rare cell type? Proteins identified with FAMA-TurboID include known interactors of this stomatal transcription factor and novel proteins that could facilitate its activator and repressor functions. Directing TurboID to stomatal nuclei enabled purification of cell type- and subcellular compartment-specific proteins. Broad tests of TurboID and miniTurbo in *Arabidopsis* and *Nicotiana benthamiana* and versatile vectors enable customization by plant researchers.

**\*For correspondence:**
bergmann@stanford.edu

## Introduction

All major processes of life, including growth and development and interactions among cells, organisms and the environment, rely on the activity and co-operation of hundreds of proteins. To fully understand these processes on a cellular level, we must know all players present in a cell or cell type at a specific location and time. This requires information about transcription and chromatin state, as well as about protein abundance and protein complex compositions. A large international effort, the 'human cell atlas' project, is taking a first step in this direction. It aims to characterize all cell types in the human body, using recent advancements in high-throughput single cell and multiplex techniques (*Regev et al., 2017*; *Stuart and Satija, 2019*). Following this example, a call for a 'plant cell atlas' describing nucleic acid, protein and metabolite composition of different cell types in plants was issued at the start of this year (*Rhee et al., 2019*). While several groups have produced single-cell gene expression profiles (e.g. *Efroni et al., 2015*; *Ryu et al., 2019*; *Denyer et al., 2019*; *Nelms and Walbot, 2019*) and tissue/cell-type-specific profiles of active translation (e.g. *Vragović et al., 2015*; *Tian et al., 2019*), we lack effective tools to obtain similarly precise information about protein distribution, abundance and the composition of protein complexes.

**eLife digest** Cells contain thousands of different proteins that work together to control processes essential for life. To fully understand how these processes work it is important to know which proteins interact with each other, and which proteins are present at specific times or in certain cellular locations. Investigating this is particularly difficult if the proteins of interest are rare, either because they are present only at low levels or because they are unique to a particular type of cell.

One such protein known as FAMA is only found in young guard cells in plants. Guard cells are rare cells that surround pores on the surface of leaves. They help open or close the pores to allow carbon dioxide and water in and out of the plant. Inside these cells, FAMA regulates the activity of genes in the nucleus, the compartment in the cell that houses the plant's DNA.

Two recently developed molecular biology tools, called TurboID and miniTurbo, allow researchers to identify proteins that are in close contact with a protein of interest or are present at a specific place inside living animal cells. These tools use a modified enzyme to add a small chemical tag to proteins that are close to it, or anything to which it is anchored. Mair et al. adapted these tools for use in plants and tested their utility in two species that are commonly used in research: a tobacco relative called *Nicotiana benthamiana,* and the thale cress *Arabidopsis thaliana*.

Their experiments showed that TurboID and miniTurbo can be used to tag proteins in different types of plant cells and organs, as well as at different stages of the plants' lives. To test whether the tools are suitable for identifying partners of rare proteins, Mair et al. used FAMA as their protein of interest. Using TurboID, they detected several proteins in close proximity to FAMA, including some that FAMA was not previously known to interact with. Mair et al. also found that TurboID could identify a number of proteins that were present in the nuclei of guard cells. This shows that the tool can be used to detect proteins in sub-compartments of rare plant cell types.

Taken together, these findings show that TurboID and miniTurbo may be customized to study plant protein interactions and to explore local protein 'neighborhoods', even for rare proteins or specific cell types. To enable other plant biology researchers to easily access the TurboID and miniTurbo toolset developed in this work, it has been added to the non-profit molecular biology repository Addgene.

Today's state-of-the-art for identification of *in-planta* protein interactors and complexes is affinity purification with protein-specific antibodies or single- and tandem affinity purification tags and subsequent mass spectrometry (MS) analysis (*Struk et al., 2019*; *Xu et al., 2010*). However, while affinity purification-mass spectrometry (AP-MS) strategies are undoubtedly useful in many instances, AP-MS is challenging for very low abundant proteins, those expressed only in rare cell types or developmental stages, and those with poor solubility like integral membrane proteins. Moreover, AP-MS tends to miss weak and transient interactions unless paired with crosslinking (*Qi and Katagiri, 2009*; *Van Leene et al., 2007*; *Bontinck et al., 2018*). For obtaining subcellular proteomes, typical traditional approaches rely on cell fractionation protocols that enrich organelles from whole tissues, followed by protein extraction and MS analysis. Besides cross-contamination with other organelles, cell fractionation has the issue that only compartments that can be purified are accessible (*Agrawal et al., 2011*). The usefulness of this strategy for studying local protein compositions in individual cell types is further limited by the requirement of prior enrichment of the cell type of interest. For rare and transient cell types, acquiring a sufficient amount of 'pure' organelle material for MS analysis would be a major challenge.

Recent technological innovations in the form of proximity labeling (PL) techniques provide the sensitivity and specificity needed to identify protein complexes and (local) proteomes on a cell-type-specific level. These techniques employ engineered enzymes to covalently attach biotin to nearby proteins which can then be affinity purified from total protein extracts using streptavidin-coupled beads without the need for crosslinking to stabilize weak and transient protein interactions or cell sorting and fractionation to enrich organelles. Because proteins do not have to be isolated in their native state, harsher extraction and more stringent wash conditions can be applied, which can reduce false positives from post-lysis interactions or from non-specific binding of proteins to the beads, and can improve solubilization of membrane proteins (for recent review see *Gingras et al.,*

*2019*). An ever growing number of applications for PL in animals, including the characterization of protein complexes (e.g. the nuclear pore complex; *Kim et al., 2014*), of organellar proteomes (e.g. mitochondrial matrix and intermembrane space; *Rhee et al., 2013*; *Hung et al., 2016*), as well as for local proteomes (e.g. inside cilia; *Mick et al., 2015*, at synaptic clefts; *Loh et al., 2016*, or at endo-plasmic reticulum-mitochondria contact sites; *Cho et al., 2017*; *Hung et al., 2017*), demonstrate the usefulness and versatility of these techniques. Although the potential utility for PL in plants is equally tremendous, its adaptation in plants has been slower, with the successes being limited primarily to systems employing transient and/or high expression. The first three of these (*Lin et al., 2017*; *Khan et al., 2018*; *Conlan et al., 2018*) utilized BioID, which is based on an *Escherichia coli* biotin ligase (BirA) made promiscuous by a point mutation (R118G) to yield BirA*. BirA* is either fused to a protein of interest or targeted to a desired subcellular localization to mark protein interactors and local proteomes, respectively. When biotin is supplied in the presence of ATP, BirA* binds and acti-vates biotin and releases reactive biotinyl-AMP which can covalently bind to close-by primary amines on lysine residues (*Roux et al., 2012*). Based on experiments with the nuclear pore complex (*Kim et al., 2014*), the labeling radius of BirA* was estimated to be approximately 10 nm, which is in the size-range of an average globular protein. The actual labeling range may vary between experi-ments and is dependent on characteristics of the BirA* fusion protein, such as linker length and bait mobility, and on the labeling time.

Structural features of plants, including cell walls and the cuticle, most plants' growth temperature optima and the fact that plants produce and store biotin in their cells, provide major impediments to BioID-based PL systems (*Bontinck et al., 2018*). Recent engineering of improved versions of BirA, however, might improve PL efficiency in plants and provide the tools required to build a 'plant cell atlas'. In a directed evolution approach, two new variants – TurboID and miniTurbo – with similar specificity as BirA* but greatly increased activity and lower temperature requirements were created (*Branon et al., 2018*). TurboID and miniTurbo were successfully utilized to generate different organ-ellar proteomes in HEK cells and work in vivo in a broad range of species, including *Drosophila* and *Caenorhabditis elegans* which were previously inaccessible for BioID (*Branon et al., 2018*). While this work was in review, a new manuscript demonstrating the effectiveness of TurboID in tobacco plants (*Zhang et al., 2019*) was published. However, TurboID was again highly expressed and the activity of miniTurbo was not tested. Moreover, the potential of TurboID to identify rare protein complexes or local proteomes in individual cell types of a complex multicellular organisms had not yet been addressed.

Here, we show that TurboID and miniTurbo enable effective PL in a wide variety of tissues and expression levels in plants. We use TurboID to identify partners of the stomatal guard cell transcrip-tion factor FAMA and to obtain the nuclear proteome of a rare cell type in *Arabidopsis* seedlings – young stomatal guard cells. Our work indicates high *sensitivity* of the new PL enzymes in plants and is complementary to recent work (*Zhang et al., 2019*) demonstrating high *specificity* in a plant-path-ogen signaling context. To enable adoption by the plant research community, we provide reagents and a broadly applicable workflow for PL experiments under different experimental conditions in a variety of tissues in *Arabidopsis* and *Nicotiana benthamiana* and highlight critical steps in experimen-tal design and execution.

## Highlights

- TurboID (TbID) and miniTurbo (mTb) work well in all tested tissues and growth stages of stably transformed *Arabidopsis* and in transiently transformed *N. benthamiana* leaves.
- Labeling times of under 10 min can give immunoblot-detectable signals, but longer incubation may be required for protein identification by mass spectrometry (MS).
- In Arabidopsis, TbID activity is higher than mTb activity, but 'background' labeling with endog-enous biotin is also increased.
- Biotin concentrations in the range of 2.5–50 µM and 20–50 µM are suitable for enhanced label-ing with TbID and mTb. For most *Arabidopsis* tissues, submergence in the biotin solution is sufficient but some tissues and other plants may require (vacuum) infiltration of the biotin solu-tion for optimal labeling.
- TbID and mTb work at temperatures compatible with normal plant growth and at elevated temperatures, but are most likely not suitable for cold stress experiments.

- Proximity labeling (PL) with the FAMA-TbID fusion protein led to the identification of new putative co-activator and -repressor complex components for FAMA, a transcription factor in young guard cells.
- PL with nuclear TbID produced general and young guard cell-specific proteomes with high specificity for nuclear proteins and identified guard cell-specific transcription factors.
- Important considerations for the experimental design of PL experiments: choice of biotin ligase (TbID vs. mTb); proper controls to distinguish specific labeling from background; optimization of labeling conditions (biotin concentration, treatment time); optimization of bead amount for affinity purification
- Important considerations for affinity purification and MS identification of biotinylated proteins: depletion of free biotin to reduce required bead amount (buffer exchange with gel filtration columns or centrifugal filters, dialysis); beads for affinity purification (avidin-, streptavidin-, or neutravidin beads vs. anti-biotin antibodies); MS sample prep (on-bead trypsin digest vs. elution and in-gel digest); MS and quantification method (label-free vs. isotopic labeling)

## Results

### TurboID and miniTurbo can biotinylate plant proteins under conditions appropriate for plant growth

With increased efficiency, BioID-based PL would be a valuable tool to study protein interactions and local proteomes on a cell-type-specific level in plants. We therefore made an initial diagnosis of whether the improved BirA* variants TurboID and miniTurbo (hereafter called TbID and mTb) are appropriate for PL applications in plants, by testing their activity in the nucleus and cytosol of two plant model systems: transiently transformed *N. benthamiana* leaves and young seedlings of stably transformed *Arabidopsis*. To enable a comparison with previous experiments in the literature, we also included the original BirA* in our experiments and expressed all three versions under the ubiquitous UBIQUITIN10 (UBQ10) promoter. A YFP tag was added to confirm correct expression and localization of the TbID and mTb biotin ligases (*Figure 1—figure supplements 1* and *2*).

To test biotin ligase activities, we treated leaf discs from *N. benthamiana* leaves or whole 5-day-old transgenic Arabidopsis seedlings, each expressing TbID or mTb, with biotin and subsequently monitored biotinylation in total protein extracts using streptavidin immunoblots. For biotin treatment, we briefly vacuum infiltrated the plant tissue with the biotin solution and then incubated the tissue submerged in the solution for 1 h at room temperature (approximately 22°C). Mock- or untreated plants were used as controls (*Figure 1A*). In TbID- and mTb-expressing *N. benthamiana* and *Arabidopsis*, biotin treatment induced strong labeling of proteins, demonstrating that the new biotin ligases work under our chosen conditions in plants. As was observed in other organisms, both TbID and mTb showed greatly increased activity compared to BirA*, which mainly achieved weak self-labeling within 1 h of biotin treatment (*Figure 1B–C*, *Figure 1—figure supplements 3* and *4*). Since plants produce and store free biotin in their cells, we were concerned about 'background' labeling in the absence of exogenous biotin. Although it did appear, background labeling was in most cases negligible. Direct comparison of TbID and mTb in our plant systems revealed little difference in either activity or background labeling in *N. benthamiana* (*Figure 1B*, *Figure 1—figure supplement 3*), possibly due to the high expression levels of the constructs. In *Arabidopsis*, however, TbID was clearly more active than mTb but also produced more background. Enhanced activity of TbID in the absence of exogenous biotin was especially evident in lines that express TbID and mTb at low levels (*Figure 1C*, *Figure 1—figure supplement 4*). Comparing nuclear and cytosolic constructs, we did not observe any significant differences in labeling efficiency at the resolution of immunoblots (*Figure 1—figure supplements 3* and *4*).

From these experiments, we conclude that both TbID and mTb are well suited for use in plants. Which version is more suitable may depend on the individual question and whether high sensitivity (TbID) or tighter control over labeling time (mTb) is important. For this current study, we generated a versatile set of gateway-compatible entry and destination vectors that can be used to express TbID or mTb alone or as fusion with a protein of interest under a promoter of choice (*Figure 1—figure supplement 5*). This 'toolbox' is accessible through Addgene (available vectors are listed in the Materials and methods section).

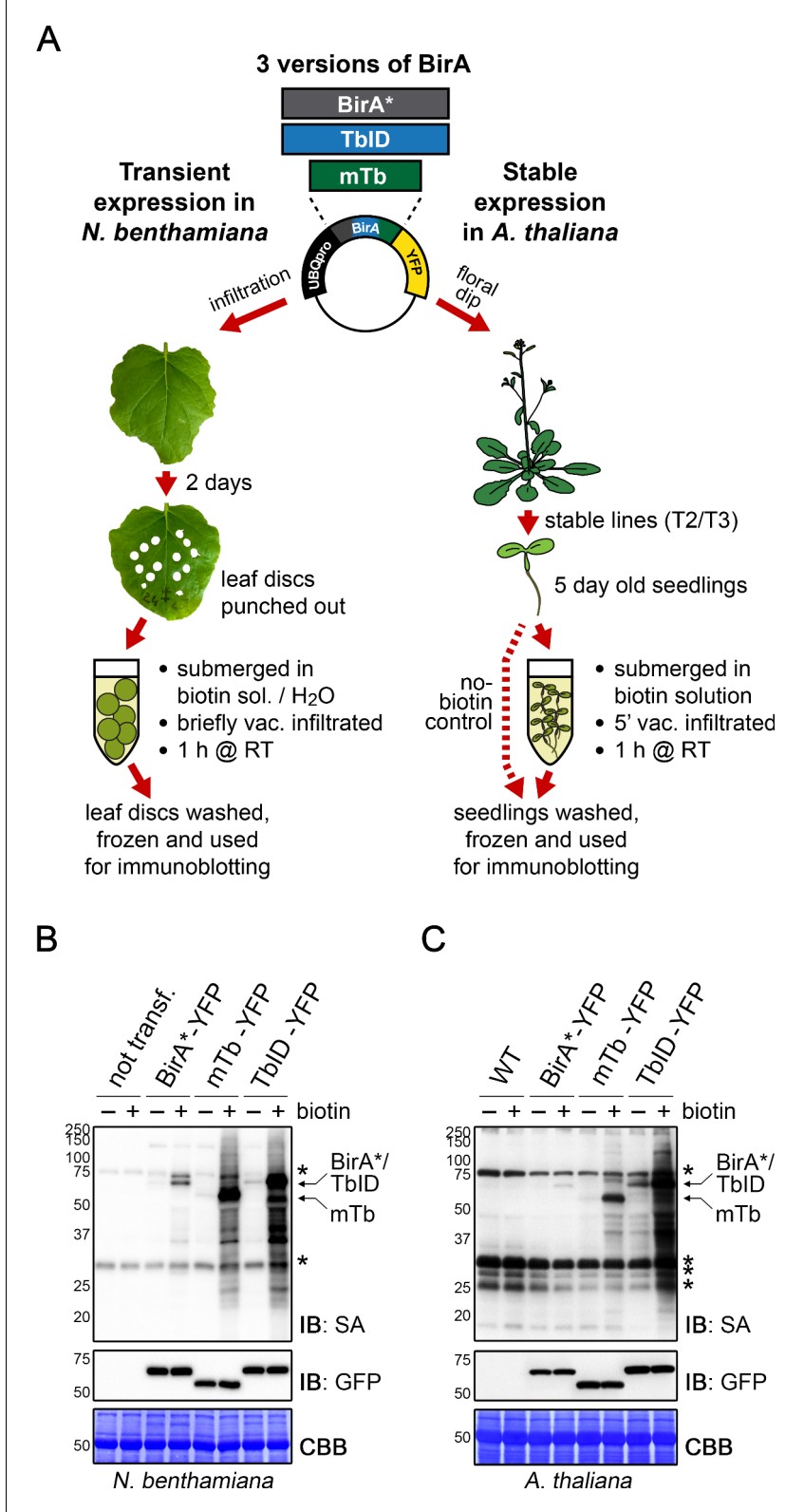

**Figure 1.** TbID and mTb exhibit robust biotinylation activity in *N. benthamiana* and *Arabidopsis*. (**A**) Overview of the experimental setup. UBQ10pro::BirA*/TbID/mTb-YFP constructs with an NLS or NES for nuclear or cytosolic localization were used for transient and stable transformation of *N. benthamiana* and *A. thaliana*, respectively. Tobacco leaf discs or whole *Arabidopsis* seedlings were submerged in a 250 µM biotin solution, briefly vacuum

*Figure 1 continued on next page*

*Figure 1 continued*

infiltrated, incubated for 1 h at room temperature (RT,~22°C) and frozen. Untreated controls were infiltrated with H₂O or frozen directly. Expression and activity of the BirA versions were analyzed by immunoblotting. (**B–C**) Biotin ligase activity in *N. benthamiana* (**B**) and *Arabidopsis* (**C**). Streptavidin (SA) and anti-GFP immunoblots (IB) of protein extracts from tobacco leaf discs and Arabidopsis expressing the cytosolic BirA variants without (-) and with (+) biotin treatment. Untransformed tobacco leaves and Col-0 wild-type (WT) seedlings were used as controls. Each sample is a pool of 3 leaf discs or ~ 30 seedlings. Coomassie Brilliant Blue-stained membranes (CBB) are shown as a loading controls. Asterisks mark the positions of naturally biotinylated proteins. For microscopy images showing the subcellular localization of the BirA variants in *N. benthamiana* and *Arabidopsis* see *Figure 1—figure supplements 1* and *2*. For immunoblots showing the activity and expression of both cytosolic and nuclear BirA versions in *N. benthamiana* and *Arabidopsis* see *Figure 1—figure supplements 3* and *4*. For a schematic overview over the generation and composition of the available vectors in the 'PL toolbox' see *Figure 1—figure supplement 5*.

The online version of this article includes the following figure supplement(s) for figure 1:

**Figure supplement 1.** Subcellular localization of biotin ligase constructs in transiently transformed *N. benthamiana* leaves.
**Figure supplement 2.** Subcellular localization of biotin ligase constructs in stable *Arabidopsis* lines.
**Figure supplement 3.** TbID and mTb are highly active in the cytosol and nucleus of transiently transformed *N. benthamiana* leaves.
**Figure supplement 4.** TbID is more active than mTb, but also produces more background labeling in *Arabidopsis*.
**Figure supplement 5.** Generating a toolbox of gateway-compatible vectors for PL in plants.

## Testing boundaries with TurboID – effects of labeling time, temperature, biotin concentration and application

Achieving an optimal enzyme efficiency by using the right experimental conditions, like labeling time, temperature, biotin concentration and mode of application, can be key for using PL with low-abundant proteins in plants. We therefore tested the effect of those parameters on biotin labeling in 4- to 5-day-old *Arabidopsis* seedlings expressing TbID and mTb under the UBQ10 promoter. In mammalian cell culture, 10 min of labeling with TbID were sufficient to visualize biotinylated proteins by immunoblot and to perform analysis of different organellar proteomes (*Branon et al., 2018*). Using immunoblots, we observed similarly fast labeling in plants. TbID induced labeling of proteins over background levels within 15–30 min of treatment with 250 or 50 µM biotin at room temperature (22°C) and labeling steadily increased over the next 3 to 5 h (*Figure 2A*, *Figure 2—figure supplement 1*). An increase in self-labeling of TbID was evident even earlier, after as little as 5 min (compare *Figure 4—figure supplement 3*). Time course experiments in *N. benthamiana* suggest that mTb is equally fast, with clear labeling of proteins visible within 10 min of treatment with 50 µM biotin (*Figure 1—figure supplement 3*). This is a significant improvement over BirA*, for which labeling times of 24 h were applied in all three published plant experiments (*Khan et al., 2018*; *Conlan et al., 2018*; *Lin et al., 2017*).

We systematically tested the effect of different biotin treatment temperatures on TbID and mTb activity in *Arabidopsis* seedlings. Encouragingly, the activity of both variants was nearly as high at 22°C as at 30°C. Moreover, TbID showed only a moderate increase of activity at 37°C, while mTb activity was actually reduced at this temperature (*Figure 2B*). High activity at ambient temperatures was also observed in *N. benthamiana* (*Figure 2—figure supplement 2*). Increasing temperatures above plant growth conditions to improve labeling is therefore not needed.

The biotin concentration used for PL is an important consideration. Endogenous levels of biotin in plants are sufficient for low-level labeling of proteins by TbID, and to some extent also by mTb. While this may be useful for some applications, most applications will require strongly enhanced and time-regulated labeling through the addition of exogenous biotin. Although using excessive amounts of biotin is inconsequential for immunoblots, it poses a problem for downstream protein purification with streptavidin beads, as will be discussed later. We therefore tested biotin concentrations ranging from 0.5 to 250 µM to determine the optimal substrate concentration for TbID and mTb. We found that TbID has a larger dynamic range than mTb. Weak over-background labeling could already be seen with 0.5 µM biotin, which increased weakly through 20 µM, followed by a

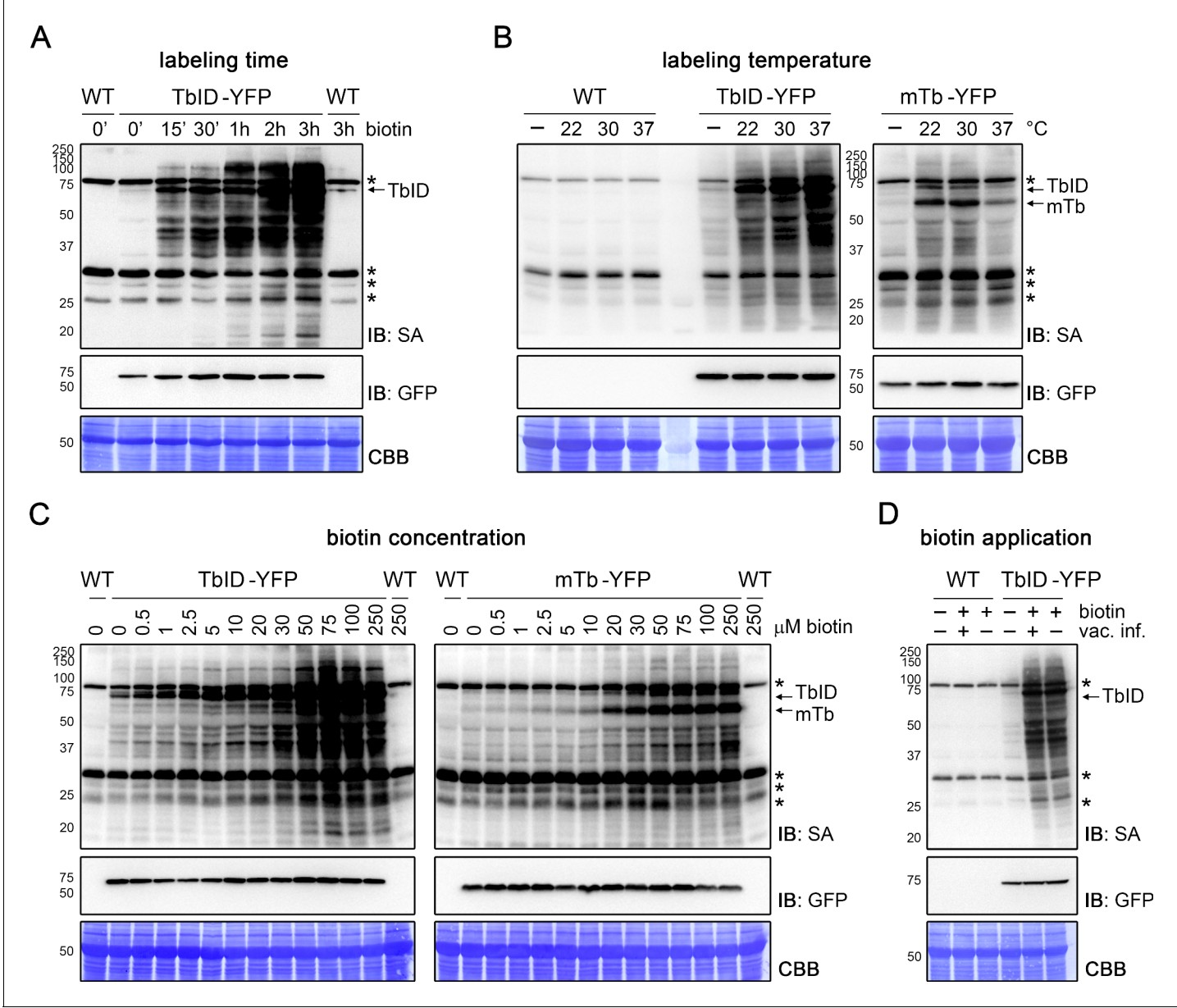

**Figure 2.** TbID and mTb work quickly and tolerate a range of experimental conditions in *Arabidopsis* seedlings. (**A–D**) Dependency of TbID and mTb activity on labeling time, temperature, biotin concentration and biotin application. Four- to 5-day-old seedlings were treated with biotin as described below. Activity and expression of the TbID/mTb-YFP constructs were analyzed by immunoblots (IB) with streptavidin-HRP (SA) and anti-GFP antibodies. Coomassie Brilliant Blue-stained membranes (CBB) are shown as a loading controls. Asterisks mark the positions of naturally biotinylated proteins. Each sample is a pool of ~30–50 seedlings. (**A**) Labeling time. Wild-type (WT) and UBQ10pro::TbID-YFP$_{NLS}$ (TbID-YFP) seedlings were submerged in 250 µM biotin, briefly vacuum infiltrated and incubated for the indicated time at room temperature (22°C). A control sample was taken before treatment (0'). (**B**) Temperature-dependency. WT, UBQ10pro::TbID-YFP$_{NLS}$ (TbID-YFP) and UBQ10pro::mTb-$_{NES}$YFP (mTb-YFP) seedlings were submerged in 250 µM biotin and incubated for 1 h at the indicated temperature. Control samples (-) were incubated in $H_2O$ at 22°C. (**C**) Biotin concentration. UBQ10pro::TbID-$_{NES}$YFP (TbID-YFP) and UBQ10pro::mTb-$_{NES}$YFP (mTb-YFP) seedlings were submerged in 0.5 to 250 µM biotin and incubated for 1 h at room temperature. A control sample was taken before treatment (0 µM). (**D**) Biotin application. WT and UBQ10pro::TbID-YFP$_{NLS}$ (TbID-YFP) seedlings were submerged in 250 µM biotin, briefly vacuum infiltrated (vac. inf.) or not and incubated for 1 h at room temperature. A control sample was taken before treatment. For a longer time course in *Arabidopsis* and quantification of the immunoblots shown in (**C**) see *Figure 2—figure supplements 1* and *3*. For a short time course and temperature dependency of TbID and mTb in *N. benthamiana* see *Figure 1—figure supplement 3* and *Figure 2—figure supplement 2*.

The online version of this article includes the following figure supplement(s) for figure 2:

**Figure supplement 1.** Biotinylation by TbID in *Arabidopsis* increases over time.

**Figure supplement 2.** TbID and mTb are active from 22°C to 37°C in *N. benthamiana*.

*Figure 2 continued on next page*

*Figure 2 continued*

**Figure supplement 3.** Quantification of TbID and mTb activity in *Arabidopsis* at different biotin concentrations.

steeper increase with 30 µM and was more or less saturated at 50–75 µM. mTb required between 2.5 and 10 µM biotin for weak activity, showed a steep increase with 20 µM (comparable to TbID) and was also saturated at 50–75 µM (*Figure 2C*, *Figure 2—figure supplement 3*). TbID and mTb are therefore comparable to BirA* in their biotin requirement and concentrations of 2.5–50 µM (TbID) and 20–50 µM (mTb) seem to be appropriate.

In initial experiments, we vacuum infiltrated the plant material with the biotin solution to maximize biotin uptake. At least in *Arabidopsis* seedlings, this is not necessary. Simply submerging the plantlets in the biotin solution resulted in the same amount of labeling as vacuum infiltration followed by incubation in the biotin solution did (*Figure 2D*). This finding is very important since it not only simplifies handling of the experiment, but also improves isolation of labeled proteins by reducing the amount of free biotin in the tissue.

## TurboID works in a wide variety of developmental stages and tissues

For TbID to be widely applicable, it must be able to biotinylate proteins in many developmental stages and plant tissues. One initial concern, especially with TbID, was that background labeling from endogenous biotin would accumulate over time, making timed experiments with older tissues unfeasible. This was, however, not the case. Labeling worked well in 4- to 14-day-old plate-grown seedlings without significant increase of background (*Figure 3—figure supplement 1*). The same was true for separated roots and shoots of 6- to 14-day-old seedlings and even for rosette leaves and flower buds of adult *Arabidopsis* plants grown on soil (*Figure 3*, *Figure 3—figure supplements 2* and *3*). Background activity was low, especially in leaf tissue, and labeling worked well. Vacuum infiltration was not required for the tested plant sample types, except for unopened floral buds, where infiltration improved labeling relative to submergence in the biotin solution (*Figure 3*). This is likely because petals and reproductive tissues are not in direct contact with the biotin solution. Overall, our experiments suggest that TbID will be applicable in a wide range of developmental stages and tissues. Since TbID and mTb behaved similar in most experiments, it is likely that the same is true for mTb.

## Testing TurboID's potential to identify partners of a very low-abundant transcription factor and to explore the nuclear proteome of a rare and transient cell type

After confirming general applicability of TbID for PL in plants, we wanted to test its performance for the identification of rare protein complexes and the characterization of cell-type-specific organellar proteomes in a real experiment. For this purpose, we chose a cell-type-specific transcription factor (FAMA) and a subcellular compartment of a rare cell type (nuclei of FAMA-expressing stomatal cells) for a case study. FAMA is a nuclear basic helix-loop-helix (bHLH) transcription factor (TF) that is expressed in young stomatal guard cells (GCs) in the epidermis of developing aerial tissues (*Ohashi-Ito and Bergmann, 2006*). The low abundance of FAMA and FAMA-expressing cells renders identification of interaction partners and cell-type-specific nuclear proteins by traditional methods challenging and makes it well suited for a proof-of-concept experiment. Potential FAMA interaction partners were previously identified in yeast-2-hybrid (Y2H) and bimolecular fluorescence complementation studies (*Chen et al., 2016*; *Kanaoka et al., 2008*; *Lee et al., 2014*; *Li et al., 2018*; *Matos et al., 2014*; *Ohashi-Ito and Bergmann, 2006*), but only few have been confirmed by in vivo functional data.

For our study, we generated plants expressing TbID and a fluorescent tag for visualization under the FAMA promoter, either as a FAMA-protein fusion or alone with a nuclear localization signal (NLS) (*Figure 4A*, *Figure 4—figure supplement 1*). By comparing proteins labeled in the FAMApro::FAMA-TbID-Venus (FAMA-TbID) and FAMApro::TbID-YFP$_{NLS}$ ($_{FAMA}$nucTbID) plants to each other and to wild-type (WT) plants and UBQ10pro::TbID-YFP$_{NLS}$ ($_{UBQ}$nucTbID) plants, we can test the ability of the system to identify (1) proteins in close proximity to FAMA (FAMA complexes), (2) the nuclear protein composition during the FAMA-cell stage, (3) the nuclear proteome in general,

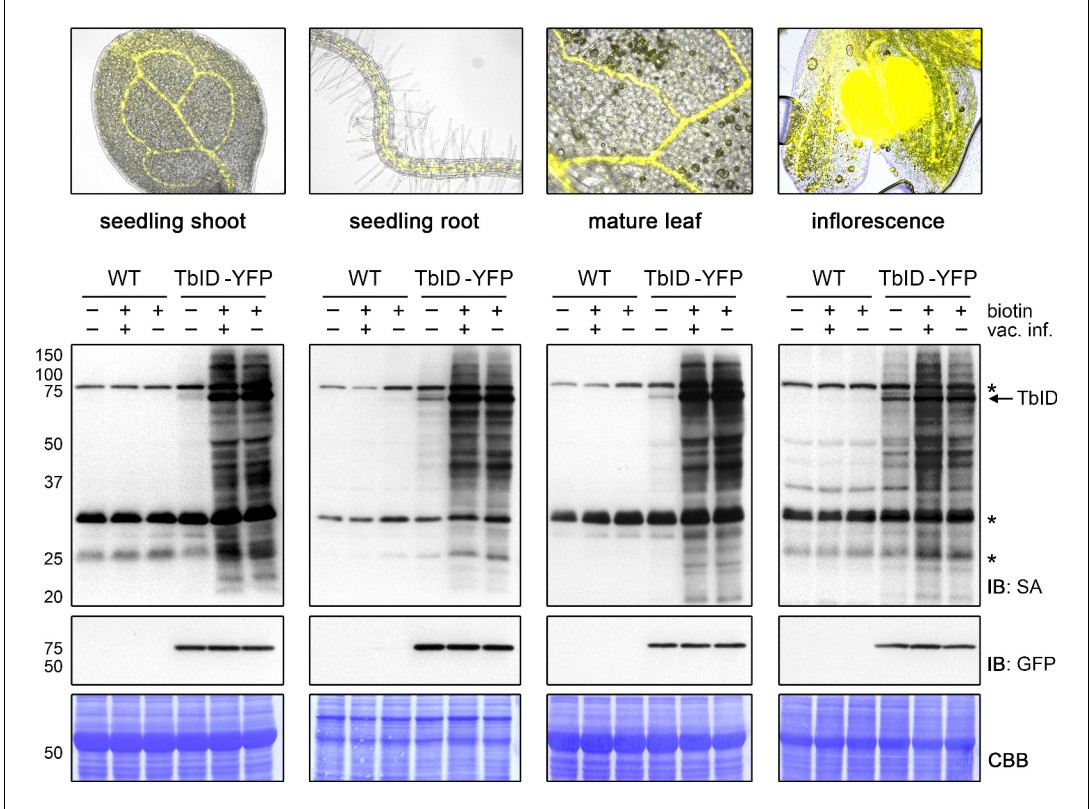

**Figure 3.** TbID works in different developmental stages and organs of Arabidopsis and does not require vacuum infiltration of biotin. TbID activity in shoots and roots of 10-day-old plate-grown UBQ10pro::TbID-YFP$_{NLS}$ (TbID-YFP) seedlings, and in rosette leaves and unopened flower buds of mature soil-grown plants. Col-0 wild-type (WT) was used as control. The plant material was submerged in a 250 μM biotin solution, briefly vacuum infiltrated until air spaces were filled with liquid or not vacuum infiltrated and incubated at room temperature (22°C) for 1 h. Control samples were taken before biotin treatment. Samples are pools of three shoots or roots, two rosette leaves or four inflorescences. Activity and expression of TbID-YFP were analyzed by immunoblots (IB) with streptavidin-HRP (SA) and anti-GFP antibodies. Coomassie Brilliant Blue-stained membranes (CBB) are shown as a loading controls. Asterisks mark the positions of naturally biotinylated proteins. Epifluorescence images of seedlings and mature tissues of the TbID-YFP line are shown on top. For immunoblots showing TbID activity and background in 4- to 14-day-old whole seedlings and shoots and roots of 6- to 14-day-old seedlings see *Figure 3—figure supplements 1* and *2*. For further microscopy images of the TbID-YFP line see *Figure 3—figure supplement 3*.

The online version of this article includes the following figure supplement(s) for figure 3:

**Figure supplement 1.** Activity and background labeling of TbID are similar in seedlings ranging from 4 to 14 days of age.

**Figure supplement 2.** Activity and background labeling of TbID are similar in roots and shoots of 6- to 14-day-old seedlings.

**Figure supplement 3.** UBQ10pro::TbID-YFP$_{NLS}$ is expressed throughout the whole plant.

and (4) possible FAMA-stage-specific nuclear proteins. The FAMA-TbID-Venus construct is functional, since it complements the seedling-lethal phenotype of the *fama-1* mutant (*Figure 4—figure supplement 2*).

Determining suitable incubation times is crucial since too short an incubation can yield insufficient protein amounts for identification but excessive incubation could label the whole subcellular compartment. We therefore performed labeling time-courses with the FAMA-TbID line, using immunoblots as a readout. FAMA auto-labeling could be observed after as little as 5 min but clear labeling of other proteins required approximately 15 to 30 min. Longer incubation led to further increase in labeling up to 3 h, both in the form of stronger discrete bands and of diffuse labeling, but stayed more or less the same thereafter (*Figure 4B*, *Figure 4—figure supplement 3*). Based on these observations, we chose 0.5 and 3 h biotin treatments, after which we would expect abundant and most FAMA interactors to be labeled, respectively, for the 'FAMA interactome' experiment (*Figure 4C*). For the 'nuclear proteome' experiment (*Figure 4D*), only the longer 3 h time point was used, since over-labeling of the compartment was not a concern. Accordingly, the FAMA-TbID,

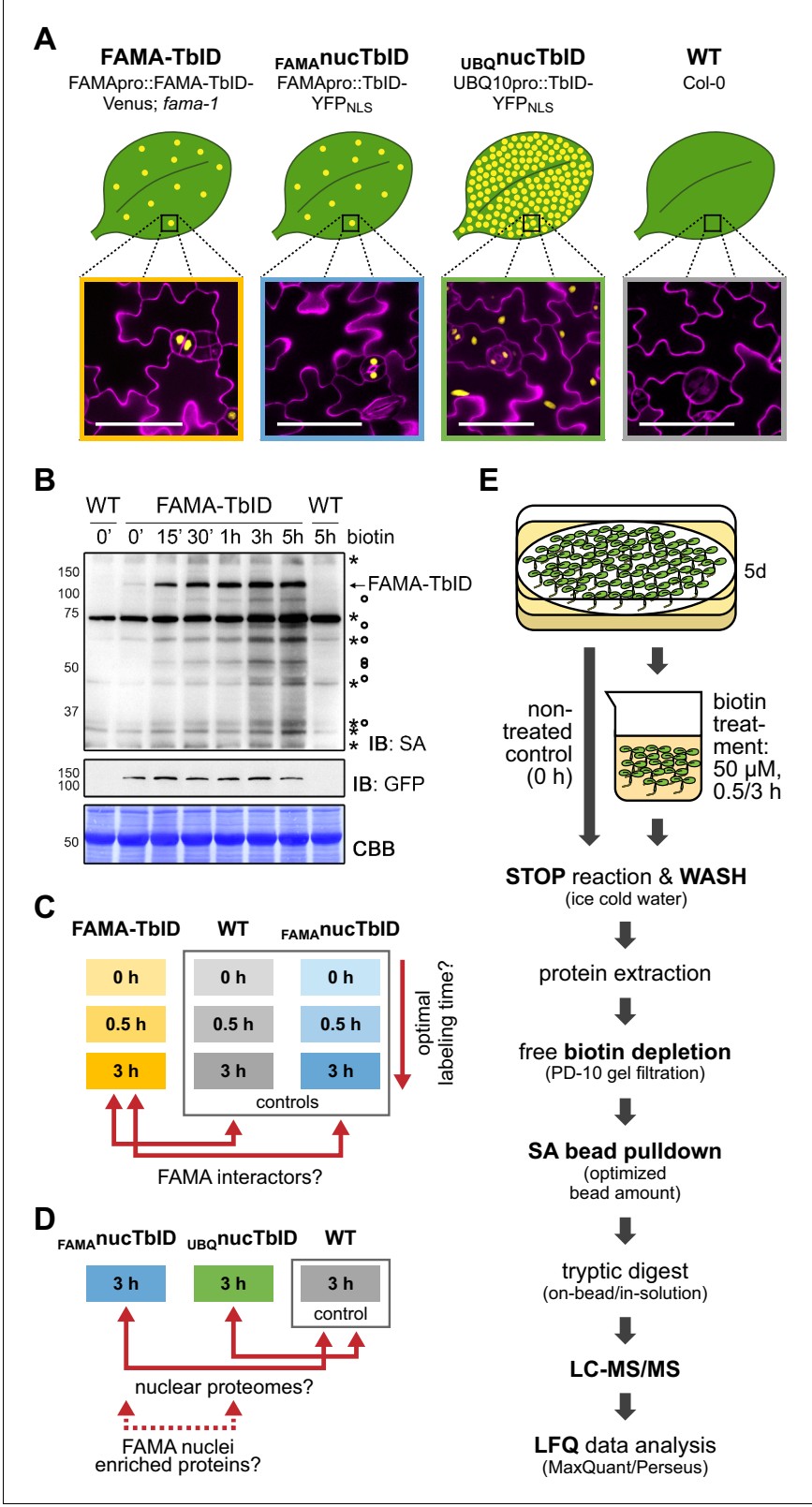

**Figure 4.** Testing TbID's potential to label protein interactors and subcellular proteomes in a rare cell type in *Arabidopsis*. (**A**) Plant lines generated for the 'FAMA interactome' and 'nuclear proteome' experiments. Line names and genotypes are given on the top, schematic expression of the TbID fusion proteins (yellow dots) in a leaf and confocal microscopy images of the epidermis of 5-day-old seedlings are shown below (TbID fusion

*Figure 4 continued on next page*

*Figure 4 continued*

protein = yellow; propidium iodide-stained cell wall = purple; scale bar = 50 µM). FAMA-TbID and $_{FAMA}$nucTbID constructs are expressed in young guard cells, while the $_{UBQ}$nucTbID construct is expressed ubiquitously. (**B**) Time course to optimize time points for the experiments. Five-day-old wild-type (WT) and FAMA-TbID seedlings were submerged in 250 µM biotin, briefly vacuum infiltrated and incubated for the indicated time at room temperature 22˚C). Control samples were taken before treatment (0'). Samples are pools of ~30 seedlings. Activity and expression of FAMA-TbID were analyzed by immunoblots (IB) with streptavidin-HRP (SA) and anti-GFP antibodies. The Coomassie Brilliant Blue-stained membrane (CBB) is shown as loading control. Asterisks and circles mark the positions of naturally biotinylated proteins and putative FAMA-TbID targets, respectively. (**C–D**) Scheme of samples and comparisons used in the 'FAMA interactome' (**C**) and 'nuclear proteome' (**D**) experiments. (**E**) Simplified workflow of the experimental procedure from biotin labeling to protein identification by liquid chromatography coupled to mass spectrometry (LC-MS/MS). Three biological replicates were used. Abbreviations: SA, streptavidin; LFQ, label free quantification. For larger extracts of the confocal microscopy images of the plant lines used in the PL experiments shown in (**A**) see *Figure 4—figure supplement 1*. For complementation of the *fama-1* phenotype by the FAMApro::FAMA-TbID-Venus construct see *Figure 4—figure supplement 2*. For another labeling time course with the FAMA-TbID line using shorter labeling times with and without vacuum infiltration of biotin see *Figure 4—figure supplement 3*. For immunoblots showing successful labeling and purification of proteins for the 'FAMA interactome' and 'nuclear proteome' experiments see *Figure 4—figure supplements 4* and *5*. For immunoblots demonstrating the importance of the biotin depletion step and a comparison of different biotin depletion strategies see *Figure 4—figure supplements 6* and *7*.
The online version of this article includes the following figure supplement(s) for figure 4:

**Figure supplement 1.** Expression of the TbID constructs in lines used for the 'FAMA interactome' and 'nuclear proteome' PL experiments.
**Figure supplement 2.** The FAMApro::FAMA-TbID-Venus construct rescues the *fama-1* mutant phenotype.
**Figure supplement 3.** FAMA-TbID self-labeling is visible within 5 min of biotin treatment and increases over time, regardless of vacuum infiltration.
**Figure supplement 4.** Confirming successful labeling of proteins in the PL experiment.
**Figure supplement 5.** Affinity purification of biotinylated proteins in the PL experiment.
**Figure supplement 6.** Free biotin from biotin treatment out-competes biotinylated proteins for streptavidin bead binding.
**Figure supplement 7.** Comparison of different biotin depletion methods and bead concentrations for an effective pulldown of biotinylated proteins.

---

$_{FAMA}$nucTbID and WT lines were treated for 0, 0.5 and 3 h, and the $_{UBQ}$nucTbID line only for 3 h. As plant material, we chose seedlings 5 days post germination, which corresponds to a peak time in FAMA promoter activity, as determined empirically by microscopy. We used three biological replicates per sample. To make the datasets as comparable as possible, all steps preceding data analysis were done together for the two experiments, as described in the next section.

## From labeling to identification of biotinylated proteins – identifying critical steps

Through empirical testing of experimental conditions using the $_{UBQ}$nucTbID line, we identified steps and choices that have a big impact on success of protein purification and identification after PL with TbID. These include sample choice to maximize bait abundance, removal of free biotin, optimizing the amount of streptavidin beads for affinity purification (AP) and choosing among MS sample prep procedures. Below, we describe our experimental procedure (*Figure 4E*) and highlight key choices.

We first labeled 5-day-old seedlings, by submerging them in a 50 µM biotin solution for 0, 0.5 or 3 hr, quickly washed them with ice cold water to stop the labeling reaction and to remove excess biotin and isolated total proteins for AP of biotinylated proteins. The protein extracts were then passed through PD-10 gel filtration columns to reduce the amount of free biotin in the sample before proceeding with AP using magnetic streptavidin beads. Successful labeling and purification was confirmed by immunoblots (*Figure 4—figure supplements 4* and *5*). The inclusion of a biotin-depletion step was found to be critical as free biotin in the protein extracts competes with biotinylated proteins for binding of the streptavidin beads (*Figure 4—figure supplement 6*). While for mammalian cell culture or rice protoplasts thorough washing of the cells seems to suffice for removal of free biotin, this is not the case for intact plant tissue (see also *Conlan et al., 2018*; *Khan et al.,*

*2018*). Especially when large amounts of starting material and moderate amounts of biotin are used, little to none of the biotinylated proteins may be bound by the beads. To maximize the amount of purified proteins it is further advisable to determine the appropriate amount of beads required for each experiment. We used 200 µl beads for approximately 16 mg total protein per sample. This amount was chosen based on tests with different bead-to-extract ratios (*Figure 4—figure supplement 7*) and was sufficient to bind most biotinylated proteins in our protein extracts, although the beads were slightly oversaturated by the highly labeled $_{UBQ}$nucTbID samples (*Figure 4—figure supplement 5D*).

Following AP, we performed liquid chromatography coupled to tandem mass spectrometry (LC-MS/MS) analysis to identify and quantify the captured proteins. Tryptic digest for LC-MS/MS analysis was done on-beads, since test experiments revealed that elution from the beads using two different methods (*Cheah and Yamada, 2017*; *Schopp and Béthune, 2018*) and subsequent in-gel digestion of biotinylated proteins yielded significantly lower protein amounts and less protein identifications (data not shown). This apparent sample loss is caused by the strong biotin-streptavidin interaction, which allows for stringent washing conditions but also prevents efficient elution of biotinylated proteins from the beads. Notably, highly biotinylated proteins, which likely comprise the most interesting candidates, will interact with more than one streptavidin molecule and will be especially hard to elute. After MS analysis, we identified and quantified the proteins by label-free quantification and filtered for significantly enriched proteins. This part was done separately for the 'FAMA interactome' and 'nuclear proteome' experiments and is described in the following sections.

## Proximity labeling is superior to AP-MS for identification of candidate interactors of FAMA

FAMA acts as both an activator and repressor for hundreds of genes (*Hachez et al., 2011*), suggesting a need for coordinated action with other TFs, co-activators and -repressors (*Matos et al., 2014*). Identifying such proteins through classical affinity purification-mass spectrometry approaches is hampered by the low overall abundance of FAMA. Apart from INDUCER OF CBF EXPRESSION 1 (ICE1), which is a known heterodimerization partner of FAMA (*Kanaoka et al., 2008*), we failed to identify any transcriptional (co-)regulators by AP-MS with FAMA-CFP, despite the use of crosslinking agents and large amounts of plant material (15 g of 4-day-old seedlings per sample). Moreover, less than 20% of the AP-MS-derived 'candidates' were predicted to be nuclear, and one quarter were chloroplast proteins (*Figure 5—figure supplement 1*, *Supplementary file 2* – Table 5). We therefore wanted to see if PL would improve the identification of biologically relevant FAMA interactors.

For the 'FAMA interactome' experiment we compared proteins purified from plants expressing the FAMA-TbID fusion (FAMA-TbID) with proteins from WT and with proteins from plants expressing nuclear TbID ($_{FAMA}$nucTbID) after 0, 0.5 and 3 h of biotin treatment. In total, we identified 2511 proteins with high confidence (quantified in all three replicates of at least one sample). Principal component analysis (PCA) showed a clear separation of the samples by genotype and time point (*Figure 5—figure supplement 2*). Despite this clear separation, the majority of proteins were common to all samples, including the untreated WT control (*Supplementary file 2* – Table 1), and moreover, were unchanged between samples (*Figure 5—figure supplements 2* and *3*). This indicates that a large proportion of identified proteins bound to the beads non-specifically. This is not uncommon for affinity purification experiments, and underlines the importance of appropriate controls and data filtering pipelines.

To remove these 'background proteins' from our dataset and to narrow down the number of FAMA complex candidates, we applied three consecutive filtering steps (*Figure 5*, for details see Materials and methods section). First, we removed proteins that were not significantly enriched in the FAMA-TbID samples compared to WT. These comprise sticky proteins that bind the beads non-specifically and a handful of proteins that are biotinylated natively by *Arabidopsis* plant biotin ligases (*Alban, 2011*; *Nikolau et al., 2003*). This was done by pair-wise comparison of FAMA-TbID and WT samples at each time point, using only proteins that were found in all three replicates of the corresponding FAMA-TbID samples, and resulted in a list of 73, 85 and 239 significantly enriched proteins (including FAMA) at the 0, 0.5 and 3 h time points, respectively (*Figure 5—figure supplement 2* – Table 2). Since TbID is highly active and endogenous levels of biotin are sufficient for low-level labeling, there is a risk that proteins are labeled stochastically and that, over time, the whole nuclear proteome would be labeled. Notably, more than half of the proteins enriched in the FAMA-TbID plants

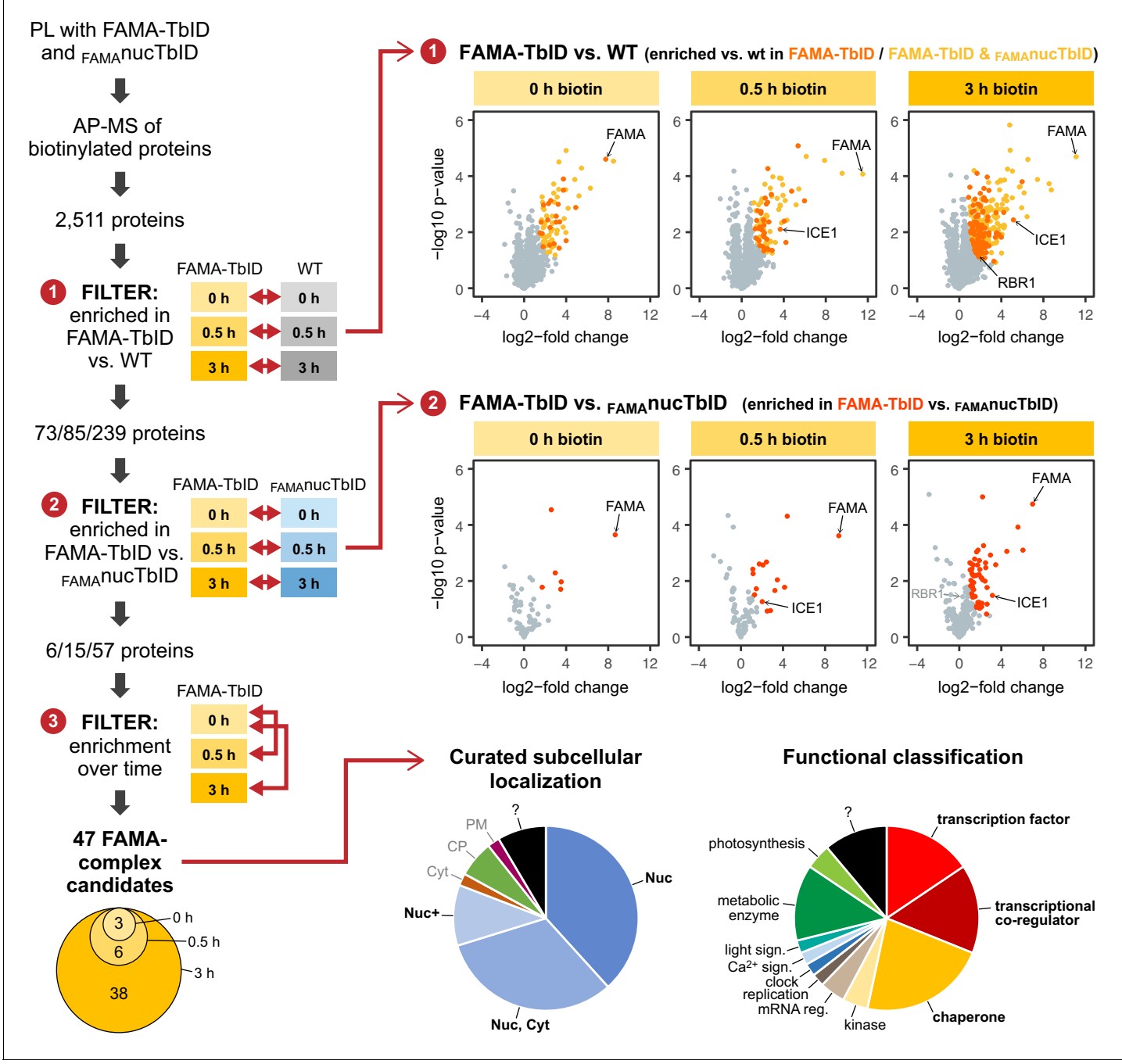

**Figure 5.** 'FAMA interactome' experiment – PL with TbID reveals potential FAMA interactors involved in transcriptional regulation. Workflow (left) and results (right) of the experimental setup and data filtering process. Biotinylated proteins from seedlings expressing FAMA-TbID or nuclear TbID in FAMA-stage cells (FAMAnucTbID) and from wild-type (WT) after 0, 0.5 and 3 h of biotin treatment were affinity purified (AP) with streptavidin beads and analyzed by mass spectrometry (MS). Proteins that were identified in all three biological replicates of at least one genotype and time point (2511 proteins) were used to filter for FAMA-complex candidates by three consecutive filtering steps: First, proteins enriched in FAMA-TbID compared to WT were determined for each time point using unpaired two-sided t-tests with a permutation-based FDR for multiple sample correction to remove background from non-specific binding to the beads. Only proteins that were identified in all three replicates of FAMA-TbID at this time point (high confidence identifications) were used for the test (cutoff: FDR = 0.05, S0 = 0.5; ❶). Significantly enriched proteins were then filtered for enrichment compared to FAMAnucTbID (same t-test parameters; ❷) to remove stochastically labeled proteins. Finally, the remaining proteins were filtered for enrichment in biotin-treated versus untreated FAMA-TbID samples (t-test: p<0.05; ❸) to remove proteins that were not labeled in response to biotin treatment. The Venn diagram on the bottom left shows the distribution of the 47 candidates between time points (see *Table 1* for candidate list). Scatter plots on the right show log2-fold changes and -log10 p-values from t-test comparisons between FAMA-TbID and WT (top) and FAMA-TbID and

*Figure 5 continued on next page*

*Figure 5 continued*

<sub>FAMA</sub>nucTbID (center). Proteins significantly enriched in FAMA-TbID are shown in yellow, orange and red. All filtering steps and statistical analyses were done in Perseus. Subcellular localization and functional distribution of the candidate proteins is shown as pie charts on the bottom right. Data were manually curated from literature. Abbreviations: Nuc, nucleus; Cyt, cytosol; CP, chloroplast; PM, plasma membrane; +, and other. For hierarchical clustering and PCA of samples used in this experiment, see *Figure 5—figure supplement 2*. For a multi scatterplot of all samples, scatterplots and heatmaps of proteins enriched in FAMA-TbID and <sub>FAMA</sub>nucTbID compared to WT and for enrichment of proteins over time in all three genotypes, see *Figure 5—figure supplements 3*, *4* and *6*, respectively. For tables summarizing all identified and enriched proteins, see *Supplementary file 2* – Tables 1-4. For an overview over the workflow and proteins identified by affinity purification of FAMA-interacting proteins using a classical AP-MS strategy with GFP-Trap beads, see *Figure 5—figure supplement 1* and *Supplementary file 2* – Table 5. For small-scale validation of selected candidates by Y2H and pairwise PL in *N. benthamiana*, see *Figure 5—figure supplement 5*.

The online version of this article includes the following figure supplement(s) for figure 5:

**Figure supplement 1.** FAMA-CFP AP-MS experiments identified ICE1, but no novel transcriptional regulators as putative FAMA partners.
**Figure supplement 2.** Clustering and PCA of samples for the 'FAMA interactome' PL experiment.
**Figure supplement 3.** Multi scatter plot of samples for the 'FAMA interactome' PL experiment.
**Figure supplement 4.** Significantly enriched proteins in the FAMA-TbID and <sub>FAMA</sub>nucTbID lines.
**Figure supplement 5.** Validation of FAMA complex candidates.
**Figure supplement 6.** Increase of biotinylation in the PL samples over time.

at any of the time points were also enriched in the <sub>FAMA</sub>nucTbID plants (*Figure 5*, *Figure 5—figure supplement 4*, *Supplementary file 2* – Table 2). We therefore applied a second filtering step to remove proteins that were not significantly enriched in the FAMA-TbID versus the <sub>FAMA</sub>nucTbID samples. Pair-wise comparison of FAMA-TbID and <sub>FAMA</sub>nucTbID samples at the three time points, further reduced the dataset to 6, 15 and 57 proteins (including FAMA) (*Figure 5—figure supplement 2*, *Supplementary file 2* – Table 2). Finally, we removed proteins that were not significantly enriched after biotin treatment compared to the untreated samples, since these proteins are likely genotype-specific contaminations. One protein, which was only enriched in the absence of exogenous biotin but not after biotin treatment, was removed as well.

This left us with 47 'high confidence' candidates (*Figure 5*, *Table 1*), 35 of which were previously demonstrated to be in the nucleus using fluorescent protein fusions or were found in MS-based nuclear proteome studies (*Figure 5*, *Table 1*, *Supplementary file 2* – Tables 2 and 4). Notably, more than half of the candidates have a role in regulation of transcription or are chaperones which could assist in FAMA's role as a TF or in protein folding and stabilization, respectively (*Figure 5*, *Table 1*, *Supplementary file 2* – Table 4). Moreover, several of these proteins have previously been shown to interact with each other, which suggests that they could be part of the same FAMA complexes. This is a huge improvement compared to our AP-MS experiment, which could only confirm FAMA's interaction with its obligate heterodimerization partner ICE1. The transcriptional regulators we found with PL can be roughly divided into two categories: TFs and transcriptional co-regulators. Among the TFs we again found ICE1 (as well as peptides shared between ICE1 and its orthologue SCRM2). We also found three other bHLH TFs (AIB/JAM1, JAM3, and BIM1) and the non-canonical bHLH-type TF BZR1. AIB and JAM3 play partially redundant roles in negative regulation of jasmonic acid (JA) signaling (*Sasaki-Sekimoto et al., 2013*; *Fonseca et al., 2014*), while BIM1 and BZR1 mediate brassinosteroid (BR) signaling (*Yin et al., 2002*; *Wang et al., 2002*). Both JA and BR signaling play roles in stomatal function or development (*Acharya and Assmann, 2009*; *Gudesblat et al., 2012*; *Kim et al., 2012*).

Among the transcriptional co-regulators we found two significantly enriched transcriptional co-activators: MED16, which is part of the mediator complex that links TFs to RNA Pol II (*Kidd et al., 2011*), and HAC1, which is a histone acetyl transferase (HAT) (*Deng et al., 2007*). Combined with previous data showing a link between FAMA and RNA Pol II (*Chen et al., 2016*), this suggests that FAMA activates genes both directly by recruiting RNA Pol II and by opening up the chromatin for other transcriptional regulators. Among transcriptional co-repressors were TOPLESS (TPL)-related proteins TPR3 and TPR4 and LEUNIG (LUG) and LEUNIG HOMOLOG (LUH), which recruit histone deacetylases (HDACs) to TFs (*Long et al., 2006*). Additionally, we identified the linker protein SEUSS (SEU), which mediates interaction of LUG and LUH with TFs (*Liu and Karmarkar, 2008*; *Sitaraman et al., 2008*). The identification of all three members of the SEU/LUG/LUH co-repressor complex is a strong indication of a functional complex with FAMA in the plant. Relaxing our filtering

**Table 1.** FAMA complex candidates from *Figure 5*.

**Enriched at time point**

| 0 h | 0.5 h | 3 h | AGI | Gene name | Functional annotation | Subcellular localization |
|---|---|---|---|---|---|---|
| | Y | Y | AT3G26744 | ICE1, SCRM | bHLH transcription factor | N |
| | Y | Y | AT2G46510 | AIB, JAM1 | bHLH transcription factor | N |
| | | Y | AT4G16430 | JAM3 | bHLH transcription factor | N, C |
| | | Y | AT5G08130 | BIM1 | bHLH transcription factor | N |
| | | Y | AT1G75080 | BZR1 | Transcription factor | N, C |
| | Y | Y | AT5G11060 | KNAT4 | Homeobox transcription factor | N, C |
| | | Y | AT2G41900 | OXS2, TCF7 | Zinc finger transcription factor | (N), C |
| | | Y | AT1G79000 | HAC1, PCAT2 | Transcriptional co-activator (histone acetyltransferase) | N, C |
| | | Y | AT4G04920 | MED16, SFR6 | Transcriptional co-activator (mediator complex) | N |
| | | Y | AT1G43850 | SEU | Transcriptional co-repressor adapter | N |
| | | Y | AT4G32551 | LUG, RON2 | Transcriptional co-repressor | N |
| | | Y | AT2G32700 | LUH, MUM1 | Transcriptional co-repressor | N |
| | | Y | AT3G15880 | TPR4, WSIP2 | Transcriptional co-repressor | N |
| | | Y | AT5G27030 | TPR3 | Transcriptional co-repressor | N |
| | | Y | AT5G02500 | HSP70-1 | HSP70 chaperone | N, C |
| | Y | Y | AT5G02490 | HSP70-2 | HSP70 chaperone | N, C |
| | | Y | AT3G09440 | HSP70-3 | HSP70 chaperone | N, C |
| | | Y | AT3G12580 | HSP70-4 | HSP70 chaperone | N, C |
| | | Y | AT5G22060 | J2 | HSP70 co-chaperone | N |
| | | Y | AT3G44110 | J3 | HSP70 co-chaperone | N, C, MA |
| | | Y | AT1G62740 | HOP2 | HSP90/70 co-chaperone | (N), C |
| | | Y | AT3G25230 | FKBP62, ROF1 | HSP90/70 co-chaperone | (N), C |
| | | Y | AT4G22670 | HIP1, TPR11 | HSP90/70 co-chaperone | N, C |
| Y | Y | Y | AT4G02450 | P23-1 | HSP90 co-chaperone | N, C |
| Y | Y | Y | AT5G56460 | | Putative protein kinase | PM |
| | | Y | AT5G35410 | SOS2, CIPK24 | Protein kinase | N, C, PM |
| | | Y | AT3G54170 | FIP37 | m6A methyltransferase complex component | N |
| | | Y | AT1G02140 | HAP1, MAGO | Exon-junction complex component | N, C |
| | | Y | AT5G41880 | POLA3, POLA4 | Putative DNA polymerase alpha subunit | N |
| | | Y | AT3G22380 | TIC | Nuclear clock regulation factor | N |
| | | Y | AT2G41100 | TCH3, CAL12 | Calcium-binding protein | N |
| | | Y | AT1G72390 | PHL | Nuclear receptor/co-activator | N, C |
| | Y | Y | AT1G20110 | FREE1, FYVE1 | ESCRT-I complex component | C, ES, N |
| | | Y | AT1G18660 | IAP1 | C3HC4-type RING-finger domain protein | MA, N |
| Y | Y | Y | AT1G12200 | FMO | Putative flavin monooxygenase | N/A |
| | Y | Y | AT3G53260 | PAL2 | Phenylalanine ammonia-lyase | N, C, EX |
| | | Y | AT3G23840 | CER26-LIKE | acyl-CoA-dependent acyltransferase | N/A |
| | | Y | AT5G13710 | CPH, SMT1 | C-24 sterol methyl transferase | N |
| | | Y | AT1G63180 | UGE3 | UDP-Glucose 4-Epimerase | C* |
| | | Y | AT5G17990 | PAT1, TRP1 | Phosphoribosy lanthranilate transferase | CP* |

*Table 1 continued on next page*

*Table 1 continued*

**Enriched at time point**

| 0 h | 0.5 h | 3 h | AGI | Gene name | Functional annotation | Subcellular localization |
|-----|-------|-----|-----|-----------|----------------------|--------------------------|
| | | Y | AT1G15980 | NDH48, NDF1 | Chloroplast NAD(P)H dehydrogenase complex subunit | CP |
| | | Y | AT4G30720 | PDE327 | Putative oxidoreductase/electron carrier | CP |
| | | Y | AT1G50570 | | Undescribed protein | N |
| | | Y | AT1G30070 | | Undescribed protein | N |
| | | Y | AT5G15680 | | Undescribed protein | N/A |
| | | Y | AT5G53330 | | Undescribed protein | N |
| | | Y | AT4G25290 | | Undescribed protein | N/A |

Column labels: Enriched at time point: time points at which a protein was significantly enriched are marked with Y. AGI: Arabidopsis gene identifier. Sub-cellular localization: as described for fluorescent protein fusions in literature unless marked with * (localization inferred from functional annotation): N, nucleus; (N), nucleus under heat or other stress; C, cytosol; EX, extracellular; PM, plasma membrane; ES, endosomes; MA, membrane (associated); CP, chloroplast; N/A, localization unknown (no experimental evidence found and localization cannot be clearly inferred from function).
For further information on the candidate proteins and selected references see **Supplementary file 2** – Table 4.

criteria to include proteins that are enriched in the FAMA-TbID vs ₍FAMA₎nucTbID samples but were not significant under our stringent cutoff, we find several more components of transcriptional co-regulator complexes, including three more MED proteins, another HAT, two more TPL-related proteins and TPL itself.

RBR1, a cell cycle regulator and a known interactor of FAMA (*Lee et al., 2014*; *Matos et al., 2014*), is also among the FAMA-TbID enriched proteins but, due to a modest fold change, did not pass our last filters (*Figure 5*, – *Supplementary file 2* Table 2). This suggests that by setting a stringent cutoff on enrichment between FAMA-TbID and ₍FAMA₎nucTbID, we might lose some true interactors. This might be especially true of ubiquitously expressed proteins with many partners like RBR1, where FAMA-RBR1 interactions are likely to represent only a small fraction of all complexes.

For a small-scale validation of candidates biotinylated by FAMA-TbID, we tested four proteins (co-repressor complex components SEU and LUH and TFs BZR1 and BIM1) for direct interaction with FAMA in a Y2H system. Additionally, we co-expressed SEU and LUH with FAMA-TbID or nuclear TbID in *N. benthamiana* and tested for biotinylation of the candidates (*Figure 5—figure supplement 5*). Our experiments suggest that BIM1 and SEU are direct interaction partners of FAMA, while LUH and BZR1 might be in indirect contact with FAMA or require a specific protein or DNA context to be present for interaction.

Overall, our 'FAMA interactome' experiment demonstrates the usefulness of PL to identify potential interaction partners of rare proteins. We identified several good FAMA-complex candidates which could support FAMA in its role as a key TF and provide a possible mechanism for FAMA to induce fate-determining and lasting transcriptional changes in developing GCs. Some of the FAMA-complex candidates identified through PL are also slightly enriched in FAMA AP-MS samples compared to their controls. However, the enrichment is not enough to call any of them, except ICE1, significant in the AP-MS experiment. PL therefore not only gave us higher specificity for nuclear proteins than the AP-MS did, but it is potentially more sensitive as well. It is worth noting, that most FAMA interaction candidates were identified at the 3 h time point and that longer biotin treatment greatly improved identification of biotinylated proteins (*Figure 5*, *Figure 5—figure supplement 4*, *Supplementary file 2* – Table 3).

## Proximity labeling can be used to analyze the nuclear proteome in rare FAMA-expressing cells during GC development

The second question our PL experiment should answer was whether TbID could be used to take a snapshot of the nuclear proteome of FAMA-expressing cells. Traditional tools to study organellar proteomes are not well-suited for such an endeavor, since they require isolation of the cell type and organelle of interest and therefore lack the required sensitivity. (*Branon et al., 2018*), showed that

TbID can be used to efficiently and specifically purify proteins from different subcellular compartments without prior cell fractionation. Their work was done using a homogeneous population of cultured mammalian cells, however, so it remained to be shown whether it would be possible to isolate an organellar proteome from an individual cell type, especially a rare or transient one, in a complex multicellular organism.

To identify nuclear proteins in FAMA-expressing young GCs and compare them to the global nuclear proteome at this growth stage, we purified proteins from seedlings expressing nuclear TbID under the FAMA (FAMAnucTbID) and UBQ10 (UBQnucTbID) promoter and from WT after three hours of biotin treatment. PCA, hierarchical clustering and multi scatterplots showed a clear separation of the three genotypes (*Figure 6—figure supplements 1* and *2*). In total, we identified 3176 proteins with high confidence (*Supplementary file 3* – Table 1). 2215 proteins were significantly enriched in UBQnucTbID compared to WT (*Figure 6*, *Figure 6—figure supplement 3*, *Supplementary file 3* – Table 2). These proteins comprise our 'global' nuclear protein dataset. Despite the relative rareness of FAMA-expressing cells, the FAMAnucTbID dataset yielded 394 proteins that were enriched compared to WT (*Figure 6*, *Figure 6—figure supplement 3*, *Supplementary file 3* – Table 3). Notably, most of them overlap with our global nuclear protein dataset (*Figure 6*), as would be expected since the UBQnucTbID dataset also contains FAMA-stage cells.

To estimate how 'pure' our nuclear proteomes are, we curated published nuclear and subnuclear compartment proteomes (*Bae et al., 2003*; *Bigeard et al., 2014*; *Calikowski et al., 2003*; *Chaki et al., 2015*; *Palm et al., 2016*; *Pendle et al., 2005*; *Sakamoto and Takagi, 2013*; *Goto et al., 2019*) and searched the *Arabidopsis* protein subcellular localization database SUBA (version 4, *Hooper et al., 2017*, http://suba.live/) for proteins that were observed in the nucleus as fluorescent-protein fusions. This resulted in a combined list of 4,681 'experimentally determined nuclear proteins'; 4021 from MS and 975 from localization studies (*Supplementary file 3* – Table 4). More than three quarters of the proteins enriched in our UBQnucTbID and FAMAnucTbID datasets are either experimentally verified nuclear proteins or are predicted to be localized in the nucleus (*Supplementary file 3* – Tables 2, 3 and 5). This suggests that most identified proteins are indeed nuclear proteins. Of the remaining proteins, most are predicted to be in the cytosol and could have been labeled by TbID right after translation and before nuclear import of the biotin ligase or by a small mis-localized fraction of TbID. Chloroplast proteins, which are a major source of contamination in plant MS experiments, make up only about 4% of our identified proteins based on experimental and prediction data (*Supplementary file 3* – Table 5). For a comparison, about 12% and 6% of the proteins identified in the two most recent *Arabidopsis* nuclear proteome studies (*Palm et al., 2016*; *Goto et al., 2019*), are predicted to be in the chloroplast (SUBAcon prediction, SUBA4). Gene ontology (GO) analysis is also consistent with nuclear enrichment in both nuclear TbID datasets (*Figure 6—figure supplement 4*, *Supplementary file 3* – Tables 2, 3 and 7). Importantly, our nuclear TbID successfully labeled all major sub-nuclear compartments and domains, including the nuclear pore complex, the nuclear envelope, the nuclear lamina, the nucleolus and other small speckles, as well as DNA- and chromatin-associated proteins (subdomain markers from *Petrovská et al., 2015*; *Tamura and Hara-Nishimura, 2013*, see *Supplementary file 3* – Table 8 for examples).

After the general assessment of data quality and nuclear specificity, we asked whether we could identify known markers for FAMA-stage GC nuclei from our dataset by comparing the FAMAnucTbID to the UBQnucTbID samples (*Figure 6*, *Figure 6—figure supplement 3*, *Supplementary file 3* – Table 3). Unlike ubiquitously expressed proteins, which should be enriched in the UBQnucTbID samples, FAMA-cell specific or highly enriched proteins should be equally abundant in both sample groups. Indeed, looking at proteins that were equally abundant or slightly enriched in the FAMAnucTbID samples, we find several known stomatal lineage- and GC-associated TFs, namely FAMA itself, HOMEODOMAIN GLABROUS 2 (HDG2) and STOMATAL CARPENTER 1 (SCAP1) (*Ohashi-Ito and Bergmann, 2006*; *Peterson et al., 2013*; *Negi et al., 2013*). Additionally, there were 10 proteins among the 44 highly FAMAnucTbID enriched proteins, that were previously identified as part of the GC proteome by Zhao and colleagues using MS analysis of 300 million GC protoplasts (*Zhao et al., 2010*; *Zhao et al., 2008*) (*Supplementary file 3* – Tables 3 and 6). It is worth noting that neither FAMA nor SCAP1 are in the Zhao GC protoplast proteome, presumably due to their relatively low expression and the use of material from more mature plants. These results suggest that PL can find lowly expressed proteins, that our knowledge of the GC proteome is not complete, and that

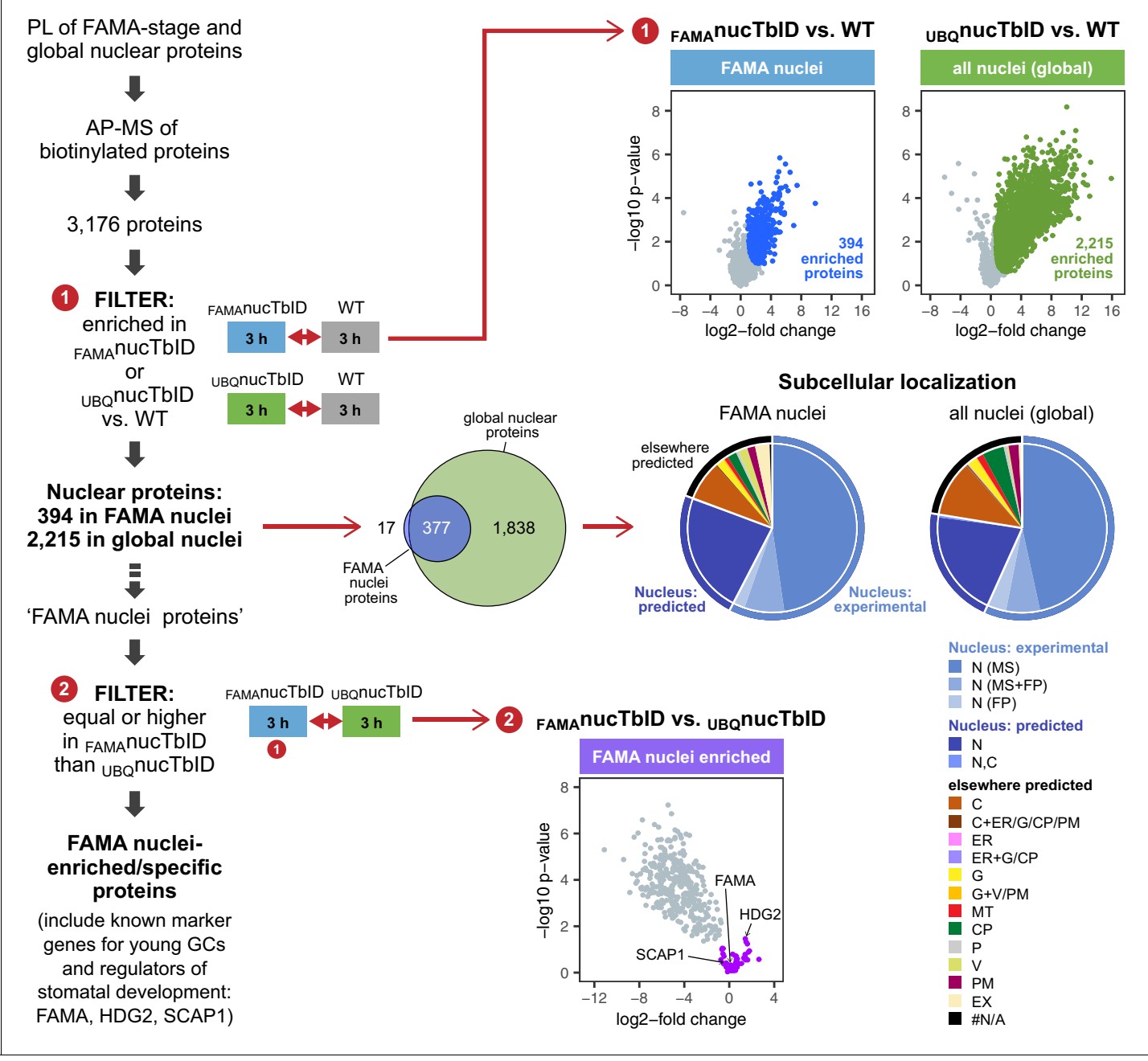

**Figure 6.** 'Nuclear proteome' experiment – PL with nuclear TbID results in identification of FAMA- and global nuclear proteomes with high organellar specificity. Workflow (left) and results (right) of the experimental setup and data filtering process. Biotinylated proteins from seedlings expressing nuclear TbID under the FAMA ($_{FAMA}$nucTbID) and UBQ10 ($_{UBQ}$nucTbID) promoters and from wild-type (WT) after 3 h of biotin treatment were affinity purified (AP) with streptavidin beads and analyzed by mass spectrometry (MS). Proteins that were identified in all three replicates of at least one genotype (3,176) were used to filter for the FAMA cell- and global nuclear proteomes: proteins enriched in $_{FAMA}$nucTbID or $_{UBQ}$nucTbID compared to WT were determined by unpaired two-sided t-tests with a permutation-based FDR for multiple sample correction. Only proteins that were identified in all three replicates of the respective nucTbID line (high confidence identifications) were used for the test (cutoff: FDR = 0.05, S0 = 0.5; ❶). Scatter plots of the log2-fold changes and -log10 p-values from the t-test comparisons are on the top right. Significantly enriched proteins are shown in blue and green. Overlap of the enriched proteins and subcellular distribution are shown as Venn diagrams and pie charts on the center right. Proteins previously found in the nucleus in MS- or fluorescent protein (FP) fusion-based experiments, and proteins predicted to be in the nucleus (SUBA4 consensus prediction) are depicted in different shades of blue (N, nucleus; C, cytosol; CP, plastid; MT, mitochondrion; ER, endoplasmic reticulum; G, golgi; P, peroxisome; V, vacuole; PM, plasma membrane; EX, extracellular; #N/A, not available). Proteins that are highly enriched in or even specific to FAMA nuclei were identified through further filtering by comparing the abundance of the 394 FAMA nuclear proteins between $_{FAMA}$nucTbID and $_{UBQ}$nucTbID using a two-sided t-test with a permutation-based FDR for multiple sample correction (FDR = 0.01, S0 = 0.1; ❷). Proteins that were not enriched in the

*Figure 6 continued*

UBQnucTbID samples (higher in FAMAnucTbID or equally abundant) were considered 'FAMA nuclei enriched/specific'. A scatterplot of the t-test results with 'FAMA nuclei enriched/specific' proteins in purple is shown at the center bottom. For hierarchical clustering and PCA of samples used in this experiment, see *Figure 6—figure supplement 1*. For multi scatterplots, scatterplots and heatmaps of proteins enriched in FAMAnucTbID and UBQnucTbID, see *Figure 6—figure supplements 2* and *3*. For a visualization of enriched GO terms, see *Figure 6—figure supplement 4*. For tables summarizing all identified and enriched proteins, published nuclear proteins and guard cell proteins, the localization data used for the pie charts in *Figure 6*, enriched GO terms of the nuclear proteomes and examples of identified proteins from different nuclear compartments, see *Supplementary file 3* – Tables 1-8.

The online version of this article includes the following figure supplement(s) for figure 6:

**Figure supplement 1.** Clustering and PCA of samples for the 'nuclear proteome' PL experiment.
**Figure supplement 2.** Multi scatter plot of samples for the 'nuclear proteome' PL experiment.
**Figure supplement 3.** Significantly enriched proteins in FAMA-expressing and all nuclei.
**Figure supplement 4.** GO terms enriched in global and FAMA nuclear proteomes.

additional important regulators of GC development and function might be uncovered by looking at stage-specific proteomes.

Overall, this experiment demonstrates the usefulness of TbID as a tool for studying subcellular proteomes on a whole-plant as well as on a cell-type-specific level. This will allow us to address questions that were previously inaccessible and thus has the potential to greatly improve our understanding of cellular processes in a cell-type-specific context.

## Discussion

### Suitability of TurboID and miniTurbo for application in plants and performance of TurboID in FAMA-complex identification and nuclear proteome analysis

Our experiments presented in this study demonstrate that the new biotin ligase versions TbID and mTb drastically improve the sensitivity of PL in plants, compared to previously used BirA*, and tolerate a range of experimental conditions. We observed rapid labeling of proteins by TbID and mTb in different species, tissues and at different growth stages from seedlings to mature plants, using a simple biotin treatment protocol at room temperature. This greatly broadens the range of possible PL applications for plants and will allow to address hitherto inaccessible or hard to address questions in the future. To test the usefulness of TbID for identification of protein interactors and organellar proteomes, we aimed to identify FAMA protein complexes and the nuclear proteome in FAMA-expressing cells. We deliberately picked a rare protein and cell type to observe its performance under conditions that make the use of traditional methods challenging or unfeasible.

In spite of FAMA's low abundance, PL with the FAMA fusion protein in our 'FAMA interactome' experiment worked very well and outperformed our FAMA AP-MS experiments both in sensitivity and in specificity for nuclear proteins. Unlike in the AP-MS experiments, we identified a number of new proteins with gene regulatory functions that could support FAMA in its role as a master regulator of GC differentiation. Beyond the known dimerization partner ICE1, which was found in both types of experiments, PL identified four additional bHLH TFs with the potential to form alterative FAMA heterodimers as well as three non-bHLH TFs. Strong, direct interaction with BIM1 could be confirmed by Y2H (*Figure 5—figure supplement 5A-B*). We further identified several epigenetic regulators, which could fulfill roles predicted for FAMA complexes based on previous genetic and transcriptomic experiments (*Adrian et al., 2015*; *Hachez et al., 2011*; *Matos et al., 2014*) and fill a gap in our current model of gene regulation during GC formation. It was satisfying to see that when FAMA complex candidates were known to form functional complexes, we often identified multiple components of the complex. One example is the SEU-LUG/LUH co-repressor complex, of which SEU acts as an adapter protein, linking LUG/LUH to TFs (*Liu and Karmarkar, 2008*). Our Y2H and PL experiments support a role of SEU as an adapter in FAMA repressive complexes (*Figure 5—figure supplement 5*). MED14, which was found as potential FAMA complex component using relaxed criteria, could also be part of this interaction chain. LUG is known to act by recruiting HDACs to promote epigenetic gene repression, but can also interact with mediator complex subunits like MED14

to interfere with their interaction with transcriptional activators (*Liu and Karmarkar, 2008*). MED14, if indeed a FAMA complex component, could therefore either interact with the FAMA heterodimer or with the repressors LUG or LUH. It is likely that our candidates are part of different activating and repressive FAMA complexes. Which proteins are in which complex and which are direct FAMA interactors will need to be further tested with independent methods.

Interestingly, we did not confirm all of FAMA's previously postulated interaction partners in our PL experiment. This has both biological and technical causes. It is possible that some previously observed interactions are too rare or conditional (e.g. MED8 may require pathogen exposure) while others probably only happen under artificial conditions like in Y2H assays (e.g. bHLH71 and bHLH93, which seem not to be required for stomatal development; *Ohashi-Ito and Bergmann, 2006*). Enrichment of NRPB3 in FAMA-TbID samples, on the other hand, could not be assessed because of non-specific binding of the protein to the beads. Other technical aspects that can complicate or prevent identification of interaction partners are removal of low-frequency interactions through stringent data filtering or a general lack of labeling. Because only proteins with exposed, deprotonated lysine residues can be labeled, sterically or chemically inaccessible proteins will not be detected.

PL of FAMA-expressing nuclei and comparison with a general nuclear proteome in our 'nuclear proteome' experiment showed that TbID is suitable to capture subcellular proteomes at a cell-type-specific level, even when the cell type is rare. We identified 2232 proteins in both nuclear datasets combined, which is 25% more than the most recent nuclear proteome obtained from cultured *Arabidopsis* cells (1528 proteins, *Goto et al., 2019*). Judging from experimentally determined and predicted localization of the identified proteins and functional annotation with GO analysis, we obtained high specificity for nuclear proteins and identified proteins from all major nuclear subdomains. The biggest 'contamination' stems from the cytosol, which could be caused by a fraction of TbID in the cytosol (e.g. right after translation) or from activated biotin diffusing out of the nucleus. Chloroplast-predicted proteins made up only a small fraction. Among FAMA-nuclei enriched proteins, we identified several known nuclear markers of young GCs, confirming our ability to detect cell-type-specific proteins, as well as proteins that have not yet been linked to GC development or function but could be interesting to investigate further. One of them is SHL (SHORT LIFE), which is a histone reader that can bind both H3K27me3 and H3K4me3 histone marks and has been implicated in seed dormancy and flower repression (*Müssig et al., 2000*; *Qian et al., 2018*). Interaction of FAMA with RBR1 and with newly identified co-repressors and co-activators from this study, strongly suggests that chromatin marks are important to lock GC in their terminally differentiated state (*Matos et al., 2014*; *Lee et al., 2014*) and SHL could be involved in this process.

## Considerations for a successful PL experiment

When designing a PL experiment, several things should be considered in order to achieve the best possible result. First, a choice has to be made which biotin ligase to use. Whether TbID or mTb is more suitable will depend on the research question. TbID is more active, which is an advantage for low abundant proteins, for cases where over-labeling is not a concern or when labeling times should be kept as short as possible. mTb is less active in the presence of endogenous levels of biotin and will give less background labeling, which could be beneficial in tissues with higher than average endogenous biotin levels (*Shellhammer and Meinke, 1990*) or when a more restricted and controlled labeling time is desired. Additional factors that should be considered when choosing a biotin ligase are whether one version works better in a specific subcellular compartment (as was the case in human HEK cells; *Branon et al., 2018*) and whether the larger TbID interferes with the activity or correct subcellular targeting of the protein of interest (POI). We added a fluorophore to all our biotin ligase constructs. This is very useful to confirm correct expression and subcellular localization of the TbID or mTb fusion protein, but may in some cases affect the activity of the ligase or the tagged protein. Interference with activity and targeting can depend on the position of the biotin ligase relative to the POI (N- or C-terminal tag). Another decision at the construct-design phase is the choice of linker length. For our experiments we added a short flexible linker to TbID. For identification of large protein complexes, increasing the linker length may improve labeling of more distal proteins.

Another crucial consideration are controls. Extensive and well-chosen controls are essential to distinguish between true candidates and proteins that either bind non-specifically to the beads or that are stochastically labeled because they are localized in the same subcellular compart as the POI. The former class of contaminants can be identified by including a non-transformed control (e.g. WT). The

best control for the latter will be situation-dependent. For identifying interaction partners of a POI, one could use free TbID or mTb targeted to the same subcellular localization as the POI, as we have done in our 'FAMA interactome' experiment. Alternatively, one or more unrelated proteins that are in the same (sub)compartment but do not interact with the POI can be used. This strategy might improve identification of proteins that are highly abundant or ubiquitously expressed but of which only a small fraction interacts with the bait. Such proteins may be lost during data filtering if a whole-compartment control is used, as we observed for RBR1 in our 'FAMA interactome' experiment. In either case, the control construct should have approximately the same expression level as the POI. For sub-organellar proteomes (e.g. a specific region at the plasma membrane) or for organellar proteomes in a specific cell type, it is useful to label the whole compartment or the compartment in all cell types for comparison.

Before doing a large-scale PL experiment, different experimental conditions should be tested to find a biotin concentration/treatment time/plant amount/bead amount combination that is suitable for the question and budget. Increasing the plant amount, labeling time or biotin concentration can improve protein coverage and can help to get more complete compartmental proteomes, but will also affect the amount of beads required. Excessive labeling is only advisable in closed compartments and when labeling of all proteins is the goal. Immunoblots are a useful tool to test different combinations of labeling concentration and time as well as for determining the correct bead amount for the pulldown so as not to over-saturate the beads. One should keep in mind, though, that the signal intensity on a western blot does not necessarily reflect the amount of labeled protein, because highly biotinylated proteins have multiple binding sites for streptavidin, which leads to signal amplification. Moreover, labeling strength will also depend on the number of sterically/chemically available sites for biotinylation, and therefore also on the size and properties of a protein.

If biotinylation is induced by addition of exogenous biotin, a crucial step before AP of biotinylated target proteins is the depletion of free biotin in the sample to reduce the amount of beads required and therefore the per sample cost. We tested the effectiveness of two different approaches: gel filtration with PD-10 desalting columns (also used by *Conlan et al., 2018*) and repeated concentration and dilution with Amicon Ultra centrifugal filters (used by *Khan et al., 2018*). The latter method has the potential to remove more biotin and to be more suitable for large amounts of plant material. In our hands, though, using Amicon centrifugal filters led to considerable sample loss, presumably due to binding of the membrane, and was very slow. PD-10 columns, in contrast, did not lead to a notable loss of biotinylated proteins (*Figure 4—figure supplement 7*). Surprisingly, consecutive filtering of protein extracts with two PD-10 columns did not improve the bead requirement. An alternative to these two methods is dialysis, which is suitable for larger volumes but is very time consuming. If the target is in an easy-to-isolate and sufficiently abundant organelle, cell fractionation prior to AP might also be considered to remove unbound biotin.

For AP, different kinds of avidin, streptavidin or neutravidin beads are available. Their strong interaction with biotin allows for efficient pulldowns and stringent wash conditions, but makes elution of the bound proteins difficult, especially if they are biotinylated at multiple sites. Should elution of the bound proteins be important for downstream processing or the identity of the biotinylated peptides be of major interest, the use of biotin antibodies might be preferable (*Udeshi et al., 2017*).

Finally, some consideration should also be given to the MS strategy. For example, digesting the proteins on the streptavidin beads instead of eluting them can increase the peptide yield. For data analysis, label-free quantification produces a more quantitative comparison of samples than comparison of peptide counts. Care should be taken that an appropriate data normalization method is chosen, especially when samples are very different, as can be the case when different cellular compartments are compared. Isotopic labeling, which allows samples to be analyzed together, can further improve quantitative comparison.

## Potential applications and challenges for PL in plants

We can see many potential applications for PL in plants, extending beyond the ones presented in this work. One that has the potential to be widely used is in vivo confirmation of suspected protein interactions or complex formation. The strategy is comparable to currently used co-immunoprecipitation (Co-IP) experiments but has the benefit that weak and transient, as well as other hard-to-purify

interactions, are easier to detect. For this approach, proteins closely related to the bait can be used as controls for interaction specificity.

One of the major applications we demonstrate in this study is de novo identification of protein interaction partners and complex components. Extending from that, PL can be used to observe changes in complex composition in response to internal or external cues (e.g. stress treatment). Currently used techniques like peptide arrays, two-hybrid screens in yeast or plant protoplasts and AP-MS have the disadvantage that they are either artificial or work poorly for low abundant and membrane proteins and tend to miss weak and transient interactions. PL could overcome some of their deficiencies. One should keep in mind, however, that rather than identifying proteins bound directly to a bait protein, PL will mark proteins in its vicinity. Labeling is generally strongest for direct interactors, but labeling radius will depend on properties of the bait such as size, mobility and linker length as well as on the duration of labeling. To define a protein interaction network, it will be useful to use several different baits (*Gingras et al., 2019*).

Another application of PL is characterization of subcellular proteomes, such as whole organellar proteomes, as we have demonstrated for the nucleus. Going forward, more detailed characterization of different organellar proteomes as well as sub-organellar proteomes and local protein composition, for example at membrane contact sites, can and should be addressed. This can be done on a whole plant level, but also at organ- and even cell-type-specific level. Importantly, PL enables investigation of previously inaccessible compartments and of rare and transient cell types as we have demonstrated for FAMA-expressing GCs. Differences between individual cell types or treatments can be investigated as well. Labeling times will, among other things, depend on the 2D or 3D mobility and distribution of the bait. For whole-compartment labeling, a combination of several baits may increase efficiency and protein coverage. One drawback of PL compared to traditional biochemical methods is that it requires the generation of transgenic plants, which limits its use to plants that can currently be transformed.

PL can also be used in combination with microscopy, to visualize the subcellular localization of biotinylated proteins and reveal labeling patterns of individual bait proteins. This can be utilized to confirm that labeling is restricted to the desired compartment, but it can also be used to fine-map the subcellular localization of a protein of interest or to obtain information about its topology, as was demonstrated for an ER transmembrane protein in human cells (*Lee et al., 2016*).

Extended uses of PL techniques that will require some modification of TbID and mTb or the PL protocol before they can be applied include interaction-dependent labeling of protein complexes (split-BioID), identification of RNAs associated with biotinylated proteins and identification of proteins associated with specific DNA or RNA sequences (for a recent review of PL methods describing these applications see *Trinkle-Mulcahy, 2019*).

While there are is a plethora of questions that can be addressed with PL, there are also limitations to what will be possible. For example, although TbID and mTb are much faster than BirA*, controlled short labeling pulses (as are possible with APEX-based PL techniques) will be hard to achieve. TbID and mTb are always active and use endogenous biotin to continuously label proteins, preventing a sharp labeling start. In addition, exogenous biotin needs time to enter the plant tissue to initiate labeling. In mammalian cell culture, 10 min of labeling might be sufficient, but in a whole multicellular organism it may take much longer, depending on the experimental setup and how complete labeling should be. In our 'nuclear proteome' experiment, for example, 3 h of biotin treatment were not sufficient to reach labeling saturation and only very few proteins were enriched in $_{FAMA}$nucTbID samples by 30 min of biotin treatment (*Figure 5—figure supplement 6*, *Supplementary file 2* – Table 3). Development of strategies to reduce background labeling, for example by (conditional) reduction of endogenous biotin levels, could improve labeling time control in the future. Another limitation stems from temperature sensitivity. Although TbID and mTb work well at room temperature and elevated temperatures, they are inactive at 4°C (*Branon et al., 2018*) and thus likely incompatible with cold treatment as might be done for cold adaptation and cold stress experiments. Further, it is likely that some compartments will be harder to work in than others. Insufficient ATP and biotin availability and adverse pH or redox conditions could reduce TbID and mTb activity. For example, although the final step of biotin synthesis happens in mitochondria (*Alban, 2011*), free biotin in mitochondria is undetectable (*Baldet et al., 1993*). It is possible that active biotin export from mitochondria will be a challenge for PL.

Going forward, it will be interesting to see how TbID and mTb perform for different applications and which challenges arise from the use in other plants, tissues and organelles. Our experiments in *Arabidopsis* and *N. benthamiana* suggest that PL will be widely applicable in plants and will provide a valuable tool for the plant community.

# Materials and methods

## Key resources table

| Reagent type (species) or resource | Designation | Source or reference | Identifiers | Additional information |
|---|---|---|---|---|
| Gene (*E. coli* - modified) | BirA* | *Branon et al., 2018*; DOI: 10.1038/nbt.4201 | | R118G mutant of BirA; promiscuous bacterial biotin ligase |
| Gene (*E. coli* - modified) | TurboID; TbID | *Branon et al., 2018*; DOI: 10.1038/nbt.4201 | | more active variant of BirA* |
| Gene (*E. coli* - modified) | miniTurbo; mTb | *Branon et al., 2018*; DOI: 10.1038/nbt.4201 | | smaller and more active variant of BirA* |
| Gene (*Arabidopsis thaliana*) | FAMA | NA | TAIR:AT3G24140 | transcription factor involved in stomatal development |
| Gene (*Arabidopsis thaliana*) | SPCH | NA | TAIR:AT5G53210 | transcription factor involved in stomatal development |
| Gene (*Arabidopsis thaliana*) | MUTE | NA | TAIR:AT3G06120 | transcription factor involved in stomatal development |
| Gene (*Arabidopsis thaliana*) | SEU | NA | TAIR:AT1G43850 | component of transcriptional co-repressor complex |
| Gene (*Arabidopsis thaliana*) | LUH | NA | TAIR:AT2G32700 | component of transcriptional co-repressor complex |
| Gene (*Arabidopsis thaliana*) | BZR1 | NA | TAIR:AT1G75080 | transcription factor involved in brassinosteroid signaling |
| Gene (*Arabidopsis thaliana*) | BIM1 | NA | TAIR:AT5G08130 | transcription factor involved in brassinosteroid signaling |
| Gene (*Arabidopsis thaliana*) | ICE1 | NA | TAIR:AT3G26744 | transcription factor involved in stomatal development and cold adaptation |
| Strain, strain background (*E. coli*) | TOP10 | other | | chemically competent *E. coli*, can be obtained from Invitrogen |
| Strain, strain background (*Saccharomyces cerevisiae*) | AH109 | Clontech | | |
| Strain, strain background (*Agrobacterium thumefaciens*) | GV3101 | other | | electrocompetent *A. thumefaciens* |
| Strain, strain background (*Nicotiana benthamiana*) | NB-1 | NA | | standard lab strain |

*Continued on next page*

*Continued*

| Reagent type (species) or resource | Designation | Source or reference | Identifiers | Additional information |
|---|---|---|---|---|
| Strain, strain background (*Arabidopsis thaliana*) | Columbia-0; Col-0 | ABRC | ABRC:CS28166 | can be obtained from ABRC |
| Genetic reagent (*Arabidopsis thaliana*) | *fama-1* | *Ohashi-Ito and Bergmann, 2006*; DOI: 10.1105/ tpc.106.046136 | ABRC:SALK_100073 | |
| Genetic reagent (*Arabidopsis thaliana*) | UBQ10pro:: BirA*-YFP$_{NLS}$ | This paper | | in Col-0 wild-type background; see Materials and methods for line generation |
| Genetic reagent (*Arabidopsis thaliana*) | UBQ10pro:: TbID-YFP$_{NLS}$ | This paper | | in Col-0 wild-type background; see Materials and methods for line generation |
| Genetic reagent (*Arabidopsis thaliana*) | UBQ10pro:: mTb-YFP$_{NLS}$ | This paper | | in Col-0 wild-type background; see Materials and methods for line generation |
| Genetic reagent (*Arabidopsis thaliana*) | UBQ10pro:: BirA*-$_{NES}$YFP | This paper | | in Col-0 wild-type background; see Materials and methods for line generation |
| Genetic reagent (*Arabidopsis thaliana*) | UBQ10pro: :TbID-$_{NES}$YFP | This paper | | in Col-0 wild-type background; see Materials and methods for line generation |
| Genetic reagent (*Arabidopsis thaliana*) | UBQ10pro:: mTb-$_{NES}$YFP | This paper | | in Col-0 wild-type background; see Materials and methods for line generation |
| Genetic reagent (*Arabidopsis thaliana*) | FAMApro::FAMA-TbID-mVenus | This paper | | in *fama-1* - /- background; see Materials and methods for line generation |
| Genetic reagent (*Arabidopsis thaliana*) | FAMApro::TbID-YFP$_{NLS}$ | This paper | | in Col-0 wild-type background; see Materials and methods for line generation |
| Genetic reagent (*Arabidopsis thaliana*) | FAMApro::FAMA-CFP | *Weimer et al., 2018*; DOI: 10.1242/dev.160671 | | |
| Genetic reagent (*Arabidopsis thaliana*) | SPCHpro::GFP$_{NLS}$ | *Adrian et al. (2015)*; DOI: 10.1016/j.devcel. 2015.01.025 | | |
| Genetic reagent (*Arabidopsis thaliana*) | MUTEpro::GFP$_{NLS}$ | *Adrian et al. (2015)*; DOI: 10.1016/j.devcel. 2015.01.025 | | |
| Antibody | Streptavidin-HRP | Thermo Fisher Scientific | Thermo Fisher Scientific:S911 | 0.2 µg/ml; 5% BSA in TBS-T |
| Antibody | Rat monoclonal anti-GFP antibody | Chromotek | Chromotek:3H9 | 1:2000; 1–5% skim milk in TBS-T |
| Antibody | Anti-HA High Affinity from rat IgG1 | Roche | Roche:11867423001 | 1:1000; 3–5% skim milk in TBS-T |
| Antibody | Myc-Tag (71D10) Rabbit mAb | Cell Signaling | Cell Signaling:2278S | 1:1000; 5% BSA in TBS-T |
| Antibody | AffiniPure Donkey Anti-Rat IgG-HRP | Jackson Immuno Research Laboratories | Jackson Immuno Research Laboratories: 712-035-153 | 1:10000; 1–5% skim milk in TBS-T |

*Continued on next page*

*Continued*

| Reagent type (species) or resource | Designation | Source or reference | Identifiers | Additional information |
|---|---|---|---|---|
| Antibody | Rabbit Anti-Rat IgG-HRP | Sigma | Sigma:A5795 | 1:10000; 1–5% skim milk in TBS-T |
| Antibody | Goat anti-Rabbit IgG (H and L), HRP conjugated | Agrisera | Agrisera:AS09 602 | 1:20000; 3–5% skim milk in TBS-T |
| Recombinant DNA reagent | R4pGWB601 (plasmid) | *Nakamura et al., 2010*; DOI: 10.1271/bbb.100184 | RIKEN BRC:pdi00133 | obtained from the Nakagawa lab (http://shimane-u.org/ nakagawa/gbv.htm) |
| Recombinant DNA reagent | R4pGWB613 (plasmid) | *Nakamura et al., 2010*; DOI: 10.1271/bbb.100184 | RIKEN BRC:pdi00099 | obtained from the Nakagawa lab (http://shimane-u.org/ nakagawa/gbv.htm) |
| Recombinant DNA reagent | R4pGWB616 (plasmid) | *Nakamura et al., 2010*; DOI: 10.1271/bbb.100184 | RIKEN BRC:pdi00102 | obtained from the Nakagawa lab (http://shimane-u.org/ nakagawa/gbv.htm) |
| Recombinant DNA reagent | pB7m34GW,0 (plasmid | *Karimi et al., 2005*; DOI: 10.1016/ j.tplants.2005.01.008 | | |
| Recombinant DNA reagent | pK7m34GW,0 (plasmid) | *Karimi et al., 2005*; DOI: 10.1016/ j.tplants.2005.01.008 | | |
| Recombinant DNA reagent | pENTR5'/TOPO (plasmid) | Invitrogen | | Gateway entry vector for promoters |
| Recombinant DNA reagent | pENTR/D-TOPO (plasmid | Invitrogen | | Gaterway entry vector for tags/genes |
| Recombinant DNA reagent | pDONR-P2R-P3 (plasmid) | Invitrogen | | Gateway entry vector for tags/genes |
| Recombinant DNA reagent | pGADT7-GW (plasmid) | *Lu et al., 2010*; DOI: 10.1111/j.1365-313X.2009.04048.x | Addgene:61702 | Gateway compatible Y2H prey vector (Gal4 activation domain) |
| Recombinant DNA reagent | pXDGATcy86 (plasmid) | *Ding et al., 2007*; DOI: 10.1385/ 1-59259-966-4:85 | | Gateway compatible Y2H bait vector (Gal4 DNA-binding domain) |
| Recombinant DNA reagent | V5-hBirA(R118G)-NES_ pCDNA3 (plasmid) | *Branon et al., 2018*; DOI: 10.1038/nbt.4201 | | obtained from Ting lab |
| Recombinant DNA reagent | V5-hBirA-Turbo-NES_ pCDNA3 (plasmid) | *Branon et al., 2018*; DOI: 10.1038/nbt.4201 | Addgene:107169 | obtained from Ting lab |
| Recombinant DNA reagent | V5-hBirA-miniTurbo-NES_pCDNA3 (plasmid) | *Branon et al., 2018*; DOI: 10.1038/nbt.4201 | Addgene:107170 | obtained from Ting lab |
| Recombinant DNA reagent | R4pGWB601_UBQ10p_ BirA(R118G)-NES-YFP (plasmid) | This paper | Addgene:127363 | UBQ10 promoter (2 kb), BirA* (cDNA) with nuclear export signal, YFP in Gateway vector R4pGWB601; see Materials and methods for cloning and Addgene for vector map |
| Recombinant DNA reagent | R4pGWB601_UBQ10p_ BirA(R118G)-YFP-NLS (plasmid) | This paper | Addgene:127365 | UBQ10 promoter (2 kb), BirA* (cDNA), YFP with nuclear import signal in Gateway vector R4pGWB601; see Materials and methods for cloning and Addgene for vector map |

*Continued*

| Reagent type (species) or resource | Designation | Source or reference | Identifiers | Additional information |
|---|---|---|---|---|
| Recombinant DNA reagent | R4pGWB601_UBQ10p_Turbo-NES-YFP (plasmid) | This paper | Addgene:127366 | UBQ10 promoter (2 kb), TurboID (cDNA) with nuclear export signal, YFP in Gateway vector R4pGWB601; see Materials and methods for cloning and Addgene for vector map |
| Recombinant DNA reagent | R4pGWB601_UBQ10p_Turbo-YFP-NLS (plasmid) | This paper | Addgene:127368 | UBQ10 promoter (2 kbA), TurboID (cDNA), YFP with nuclear import signal in Gateway vector R4pGWB601; see Materials and methods for cloning and Addgene for vector map |
| Recombinant DNA reagent | R4pGWB601_UBQ10p_miniTurbo-NES-YFP (plasmid) | This paper | Addgene:127369 | UBQ10 promoter (2 kb), miniTurbo (cDNA) with nuclear export signal, YFP in Gateway vector R4pGWB601; see Materials and methods for cloning and Addgene for vector map |
| Recombinant DNA reagent | R4pGWB601_UBQ10p_miniTurbo-YFP-NLS (plasmid) | This paper | Addgene:127370 | UBQ10 promoter (2 kb), miniTurbo (cDNA), YFP with nuclear import signal in Gateway vector R4pGWB601; see Materials and methods for cloning and Addgene for vector map |
| Recombinant DNA reagent | pB7m34GW,0_FAMAp_gFAMA-Turbo-Venus (plasmid) | This paper | | FAMA promoter (2.4 kb), FAMA (genomic DNA), TurboID, Venus in Gateway vector pB7m34GW,0; see Materials and methods for cloning |
| Recombinant DNA reagent | R4pGWB601_FAMAp_Turbo-YFP-NLS (plasmid) | This paper | | FAMA promoter (2.4 kb), TurboID, YFP with nuclear import signal in Gateway vector R4pGWB601; see Materials and methods for cloning |
| Recombinant DNA reagent | pK7m34GW,0_UBQ10p_cFAMA-TbID-Venus (plasmid) | This paper | | UBQ10 promoter (2 kb), FAMA (cDNA), TurboID, Venus in Gateway vector pB7m34GW,0; see Materials and methods for cloning |
| Recombinant DNA reagent | R4pGWB613_UBQ10p_ICE1-3xHA (plasmid) | This paper | | UBQ10 promoter (2 kb), ICE1 (cDNA) in Gateway vector R4pGWB613; see Materials and methods for cloning |
| Recombinant DNA reagent | R4pGWB613_UBQ10p_MUTE-3xHA (plasmid) | This paper | | UBQ10 promoter (2 kb), MUTE (cDNA) in Gateway vector R4pGWB616; see Materials and methods for cloning |
| Recombinant DNA reagent | R4pGWB313_UBQ10p_SEU-4xmyc (plasmid) | This paper | | UBQ10 promoter (2 kb), SEU (cDNA) in Gateway vector R4pGWB616; see Materials and methods for cloning |

*Continued on next page*

*Continued*

| Reagent type (species) or resource | Designation | Source or reference | Identifiers | Additional information |
|---|---|---|---|---|
| Recombinant DNA reagent | R4pGWB613_UBQ10p _LUH-3xHA (plasmid) | This paper | | UBQ10 promoter (2 kb), LUH (cDNA) in Gateway vector R4pGWB613; see Materials and methods for cloning |
| Recombinant DNA reagent | pXDGATcy86-FAMA (plasmid) | This paper | | FAMA (cDNA) in Gateway compatible Y2H bait vector pXDGATcy86; see Materials and methods for cloning |
| Recombinant DNA reagent | pGADT7-GW-ICE1 (plasmid) | This paper | | ICE1 (cDNA) in Gateway compatible Y2H prey vector pGADT7-GW; see Materials and methods for cloning |
| Recombinant DNA reagent | pGADT7-GW-MUTE (plasmid) | This paper | | MUTE (cDNA) in Gateway compatible Y2H prey vector pGADT7-GW; see Materials and methods for cloning |
| Recombinant DNA reagent | pGADT7-GW-SEU (plasmid) | This paper | | SEU (cDNA) in Gateway compatible Y2H prey vector pGADT7-GW; see Materials and methods for cloning |
| Recombinant DNA reagent | pGADT7-GW-LUH (plasmid) | This paper | | LUH (cDNA) in Gateway compatible Y2H prey vector pGADT7-GW; see Materials and methods for cloning |
| Recombinant DNA reagent | pGADT7-GW-BZR1 (plasmid) | This paper | | BZR1 (cDNA) in Gateway compatible Y2H prey vector pGADT7-GW; see Materials and methods for cloning |
| Recombinant DNA reagent | pGADT7-GW-BIM1 (plasmid) | This paper | | BIM1 (cDNA) in Gateway compatible Y2H prey vector pGADT7-GW; see Materials and methods for cloning |
| Recombinant DNA reagent | Additional plasmids | | | for list of Gateway compatible vectors to generate N- and C-terminal fusions with TbID or mTb ('PL toolbox') see table in Materials and methods section |
| Sequence-based reagent | Primers | | | see primer table in Materials and methods |
| Peptide, recombinant protein | Biotin powder | Sigma | Sigma:B4639 | |
| Commercial assay or kit | BioRad protein assay | BioRad | BioRad:5000006 | |
| Commercial assay or kit | Novex Colloidal blue staining kit | Invitrogen | Invitrogen: LC6025 | |
| Chemical compound, drug | Dynabeads MyOne Streptavidin C1 | Invitrogen | Thermo Fisher:65002 | |
| Chemical compound, drug | Dynabeads MyOne Streptavidin T1 | Invitrogen | Thermo Fisher:65601 | |
| Chemical compound, drug | GFP-Trap_MA beads | ChromotTek | ChromoTek:gtma-20 | |

*Continued on next page*

*Continued*

| Reagent type (species) or resource | Designation | Source or reference | Identifiers | Additional information |
|---|---|---|---|---|
| Software, algorithm | MaxQuant | *Tyanova et al., 2016a*; DOI: 10.1038/nprot.2016.136 | | version 1.6.2.6 |
| Software, algorithm | Perseus | *Tyanova et al., 2016b*; DOI: 10.1038/nmeth.3901 | | version 1.6.2.3 |
| Software, algorithm | Normalyzer | *Chawade et al., 2014*; DOI: 10.1021/pr401264n | | version 1.1.1.1 (web interface: http://normalyzer. immunoprot.lth.se/) |
| Software, algorithm | R studio | *RStudio Team, 2016* | | |
| Software, algorithm | SUBA4 | *Hooper et al., 2017*; DOI: 10.1093/nar/gkw1041 | | web interface: http://suba.live/ |
| Software, algorithm | AgriGO v2 | *Tian et al., 2017*; DOI: 10.1093/nar/gkx382 | | web interface: http://systemsbiology. cau.edu.cn/agriGOv2/ |
| Software, algorithm | REViGO | *Supek et al., 2011*; DOI: 10.1371/journal. pone.0021800 | | web interface: http://revigo.irb.hr/ |
| Other | PD-10 Sesalting Column | GE-Healthcare | Fisher Scientific: 45-000-148 | |

## Generation of the 'PL toolbox' vectors

### Cloning of gateway-compatible entry vectors containing different BirA variants

BirA* (R118G), TurboID (TbID) or miniTurbo (mTb) were amplified from V5-hBirA(R118G)-NES_pCDNA3, V5-hBirA-Turbo-NES_pCDNA3 or V5-hBirA-miniTurbo-NES_pCDNA3 (*Branon et al., 2018*) using primers BirA-fw and BirA-rv or BirA-fw and BirA-NES-rv, thereby retaining the N-terminal V5 tag, either removing or retaining the C-terminal NES and adding a GGGGSGGG linker and an AscI restriction site to both ends. The TbID and mTb PCR products were then cloned into a pENTR vector containing YFP between the attL1 and L2 sites to generate YFP-TbID and YFP-mTb fusions or into a pDONR-P2R-P3 vector containing mVenus-STOP between the attR2 and L3 sites to generate TbID-mVenus-STOP and mTb-mVenus-STOP fusions by restriction cloning with AscI. The AscI site in the pDONR-P2R-P3 vector was first introduced by mutagenesis PCR using primers pDONR-mut-AscI-fw and pDONR-mut-AscI-rv. These vectors can be used for three-way gateway recombination with a promoter and gene of choice as described below. For two-way recombination with a promoter of choice, TbID and mTb (with and without NES) were further amplified from the pDONR-P2R-P3 plasmids described above using primers BirA-YFPnls-fw and BirA-YFPnls-rv to remove the N-terminal linker and to add a NotI and NcoI site at the N-and C-terminus, respectively. BirA* was directly amplified from the PCR products with BirA-fw and BirA-rv or BirA-NES-rv in the same way. The resulting PCR products were either cloned into a pENTR vector containing YFP followed by a stop codon (versions with NES) or a YFP followed by an NLS and a stop codon (versions without NES) between the attL1 and L2 sites using NotI and NcoI to generate BirA-$_{NES}$YFP and BirA-YFP$_{NLS}$ fusions (BirA = BirA*, TbID or mTb). For a schematic overview over the cloning process and the composition of the vectors, see *Figure 1—figure supplement 5*. For a list of vectors for three- and two-way recombination see the 'PL toolbox' vectors table at the end of the Materials and methods section.

### Cloning of binary vectors for proximity labeling

The UBQ10pro::BirA-YFP$_{NLS}$ (BirA = BirA*, TbID or mTb) and FAMApro::TbID-YFP$_{NLS}$ constructs were generated by LR recombination of the R4pGWB601 backbone (*Nakamura et al., 2010*) with either a pENTR5'/TOPO containing the 2 kb UBQ10 promoter or a pJET containing 2.4 kb of the FAMA (AT3G24140) upstream sequence (positions −2420 to −1, flanked by attL4 and R1, cloned from pENTR/D-TOPO (*Ohashi-Ito and Bergmann, 2006*) by changing attL1 and L2 to L4 and R2)

and the pENTR-BirA-YFP$_{NLS}$ plasmids described above. The UBQ10pro::$_{NES}$YFP-BirA constructs were generated by LR recombination of the R4pGWB601 backbone with a pENTR5′/TOPO containing the 2 kb UBQ10 promoter and the pENTR-BirA-$_{NES}$YFP plasmids described above. For a schematic overview over the cloning process and composition of the vectors, see *Figure 1—figure supplement 5*. For a list of vectors, see 'PL toolbox' vectors table at the end of the Materials and methods section. The FAMApro::FAMA-TbID-Venus construct was generated by LR recombination of the pB7m34GW,0 backbone (*Karimi et al., 2005*) with the pJET containing 2.4 kb of the FAMA upstream sequence, with pENTR containing the genomic sequence of FAMA without stop codon flanked by attL1 and L2 (amplified from genomic DNA with primers gFAMA-fw and gFAMA-rv and recombined with pENTR/D-TOPO) and with pDONR-P2R-P3-TbID-mVenus (described above).

### Cloning of FAMA interaction partner candidates for Y2H and PL assays in *N. benthamiana*

SEU (AT1G43850), LUH (AT2G32700), BZR1 (AT1G75080), BIM1 (AT5G08130), ICE1 (AT3G26744) and MUTE (AT3G06120) were amplified from Col-0 cDNA with the primers listed in the primer table and cloned into pENTR/D-TOPO. FAMA in pENTR was cloned as described in *Ohashi-Ito and Bergmann (2006)*. For Y2H assays, FAMA was recombined with Gateway compatible pXDGATcy86 (*Ding et al., 2007*), containing the Gal4 DNA-binding domain, and SEU, LUH, BZR1, BIM1, ICE1 and MUTE were recombined into Gateway compatible pGADT7-GW (*Lu et al., 2010*), containing the Gal4 activation domain. For PL assays in tobacco, pENTR5′/TOPO containing the 2 kb UBQ10 promoter, pENTR containing FAMA cDNA and pDONR-P2R-P3-TbID-mVenus were recombined with pK7m34GW,0 (*Karimi et al., 2005*) to generare UBQ10pro::cFAMA-TbID-mVenus. Entry vectors carrying ICE1, MUTE and LUH and the entry vector carrying SEU were recombined with pENTR5′/TOPO containing the 2 kb UBQ10 promoter and R4pGWB613 (3x HA tag) and R4pGWB616 (4x myc tag), respectively (*Nakamura et al., 2010*). UBQ10pro::TbID-YFP$_{NLS}$ in R4pGWB601 is described above.

### Transformation and biotin activity assays in *N. benthamiana*

*N. benthamiana* ecotype NB-1 was transformed with UBQ10pro::BirA-YFP$_{NLS}$ and UBQ10pro::BirA-$_{NES}$YFP (BirA = BirA*, TbID or mTb) by infiltrating young leaves with a suspensions of Agrobacteria (strain GV3101) carrying one of the binary vectors. Agrobacteria were grown from an overnight culture for 2 h, supplemented with 150 µM Acetosyringone, grown for another 4 h, pelleted and resuspended in 5% sucrose to an OD600 of 2. For more stable expression, Agrobacteria carrying a 35S::p19 plasmid (tomato bushy stunt virus (TBSV) protein p19) were co-infiltrated at a ratio of 1:1 for the temperature-dependency experiment. Two days after infiltration, expression was confirmed by epifluorescence microscopy and 5 mm wide leaf discs were harvested. Two to three discs were combined per sample. They were submerged in a 50 or 250 µM biotin solution, quickly vacuum infiltrated until the air spaces were filled with liquid and incubated at the indicated temperature for 1 h. Control samples were not treated or were infiltrated with H$_2$O. After biotin treatment, leaf discs were dried and flash-frozen for later immunoblotting. All experiments were done in duplicates with leaf discs for each of the two replicates taken from different plants if possible. Only one replicate is shown. Activity of different BirA variants was compared in four independent experiments, with similar results, temperature dependency of TbID and mTb was tested in two and one experiment, respectively.

### *Arabidopsis* lines used in this study

*Arabidopsis thaliana* Col-0 was used as wild-type (WT). The *fama-1* mutant line is SALK_100073 (*Ohashi-Ito and Bergmann, 2006*). Plant lines for testing the activity of BirA*, TbID and mTb (UBQ10pro::BirA*-YFP$_{NLS}$, UBQ10pro::TbID-YFP$_{NLS}$, UBQ10pro::mTb-YFP$_{NLS}$, UBQ10pro::BirA*-$_{NES}$YFP, UBQ10pro::TbID-$_{NES}$YFP, UBQ10pro::mTb-$_{NES}$YFP) and for the 'FAMA interactome' and nuclear proteome' experiments (FAMApro::FAMA-TbID-mVenus in *fama-1*, FAMApro::TbID-YFP$_{NLS}$) were generated by floral dip of WT or *fama-1* +/- plants with the plasmids described above using agrobacterium strain GV3101. Selection was done by genotyping PCR (*fama-1*) and segregation analysis. We did not observe any obvious decrease in viability or developmental delay in our

transgenic Arabidopsis plants. All lines had a single insertion event and were either heterozygous T2 or homozygous T3 or T4 lines. While screening for biotin ligase lines with the UBQ10 promoter, we observed that most regenerants had very weak YFP signal, especially the nuclear constructs. This was not observed with any of the cell-type-specific promoters we tested. Lines used for the FAMA-CFP AP-MS experiments were previously described in other studies: FAMApro::FAMA-CFP (*Weimer et al., 2018*), SPCHpro::GFP$_{NLS}$ and MUTEpro::GFP$_{NLS}$ (*Adrian et al., 2015*).

## Plant growth conditions and biotin assays in *Arabidopsis*

Seeds were surface sterilized with ethanol or bleach and stratified for 2 to 3 days. For biotin treatment in whole *Arabidopsis* seedlings or roots and shoots, seedlings were grown on ½ Murashige and Skoog (MS, Caisson labs) plates containing 0.5% sucrose for 4 to 14 days under long-day conditions (16 h light/8 h dark, 22°C). For treatment of rosette leaves and flowers, seedlings were transferred to soil and grown in a long-day chamber (22°C) until the first flowers emerged, at which point medium sized rosette leaves (growing but almost fully expanded) and inflorescences with unopened flower buds were harvested. All samples were pools from several individual plants. Biotin treatment was done by submerging the plant material in a biotin solution (0.5–250 µM biotin in water) and either vacuum infiltrating the tissue briefly until the air spaces were filled with liquid (approximately 5 min) or not, followed by incubation at room temperature (22°C), 30°C or 37°C for up to 5 h. Controls were either treated with H$_2$O or not treated. Following treatment, the plant material was dried and flash-frozen for later immunoblotting. To confirm reproducibility of the experiment, most experiments were done in duplicates (only one replicate shown) or repeated more than once with similar results. Comparison of all different BirA variants in Arabidopsis was done once with two independent lines for each NLS and NES constructs. Difference in activity and background between TbID and mTb matches other comparisons done with varying temperature and biotin concentration. Comparison of TbID and mTb at different temperatures was done three times. Comparison of the biotin ligase activity with different biotin concentrations was done twice for TbID and once for mTb. Three time courses with up to three or five hours of biotin treatment were done with the UBQ10pro::TbID-YFP$_{NLS}$ line and time courses with the FAMA-TbID line were done in duplicates. Experiments testing the effect of biotin application with and without vacuum infiltration in different tissues were done in duplicates. Activity of TbID in 4- to 14-day-old seedlings was tested in two independent experiments. Activity of TbID in roots and shoots of 6- to 14-day-old seedlings was tested once.

## Optimization of streptavidin (SA) pulldown conditions

### Saturation of SA beads by free biotin

To test the impact of free biotin from biotin treatment on the affinity purification (AP) efficiency with SA-coupled beads, WT and UBQ10pro::TbID-YFP$_{NLS}$ seedlings were submerged in H$_2$O or 50 µM biotin for 1 h, washed twice and frozen. AP was done in essence as described in *Schopp et al. (2017)*. Briefly, 1 ml of finely ground plant material was resuspended in an equal volume of ice cold extraction buffer (50 mM Tris pH 7.5, 500 mM NaCl, 0.4% SDS, 5 mM EDTA, 1 mM DTT, 1 mM PMSF and 1x complete proteasome inhibitor), sonicated in an ice bath four times for 30 s on high setting using a Bioruptor UCD-200 (Diagenode) with 1.5 min breaks on ice and supplemented with Triton-X-100 to reach a final concentration of 2%. 2.3 ml of ice cold 50 mM Tris pH 7.5 were added to dilute the extraction buffer to 150 mM NaCl and samples were centrifuged at top speed for 15 min. The supernatant was mixed with 50 µl Dynabeads MyOne Streptavidin T1 (Invitrogen) that were pre-washed with equilibration buffer (50 mM Tris pH 7.5, 150 mM NaCl, 0.05% Triton-X-100, 1 mM DTT) and incubated over night at 4°C on a rotor wheel. The beads were washed twice each with wash buffer 1 (2% SDS), wash buffer 2 (50 mM Hepes pH 7.5, 500 mM NaCl, 1 mM EDTA, 1% Triton-X-100, 0.5% Na-deoxycholate), wash buffer 3 (10 mM Tris pH 8, 250 mM LiCl, 1 mM EDTA, 0.5% NP-40, 0.5% Na-deoxycholate) and wash buffer 4 (50 mM Tris pH 7.5, 50 mM NaCl, 0.1% NP-40). Protein were eluted by boiling the beads for 15 min at 98°C in elution buffer (10 mM Tris pH 7.5, 2% SDS, 5% beta mercaptoethanol, 2 mM biotin) and used for immunoblotting and SDS-PAGE. Two experiments with similar results were done.

## Testing of biotin depletion strategies and optimization of the SA bead amount

To compare different biotin depletion strategies and determine the required bead amount for the PL experiments, 5-day-old UBQ10pro::TbID-YFP$_{NLS}$ seedlings were submerged in a 50 µM biotin solution for 3 h, washed three times with ice cold water, frozen and later used for biotin depletion and AP experiments. Proteins were extracted as described later in 'Affinity purification of biotinylated proteins'. For biotin depletion with one or two PD-10 gel filtration columns (GE-Healthcare), the columns were equilibrated according to the manufacturer's instructions with extraction buffer without PMSF and complete protease inhibitor (50 mM Tris pH 7.5, 150 mM NaCl, 0.1% SDS, 1% Triton-X-100, 0.5% Na-deoxycholate, 1 mM EGTA, 1 mM DTT). 2.5 ml protein extract were loaded ono the column. For the trial with one PD-10 column, the gravity protocol was used, eluting with 3.5 ml equilibration buffer. For the trial with two PD-10 columns, the proteins were eluted from the first column with 2.5 ml equilibration buffer using the spin protocol. The flow through was applied to a second PD-10 column and eluted with 3.5 ml equilibration buffer using the gravity protocol. For biotin depletion with an Amicon Ultra-4 Centrifugal filter (Millipore Sigma), 625 µl protein extract were concentrated three times to 10–20% of the starting volume by centrifuging in a swinging bucket rotor at 4,000 g and 4°C and diluted with extraction buffer to the initial volume. To test bead requirements, a volume equivalent to 1/5 of the amount used in the 'FAMA interactome' and 'nuclear proteome' PL experiments was incubated with 5 to 30 µl Dynabeads MyOne Streptavidin C1 (Invitrogen) over night at 4°C on a rotor wheel. The beads were either washed once with 1 M KCl and with 100 mM Na$_2$CO$_3$ and twice with extraction buffer (1x PD-10) or just twice with extraction buffer (Amicon Ultra, 2x PD-10) before elution of bound proteins by boiling for 10 min at 95°C in 4x Leammli buffer supplemented with 20 mM DTT and 2 mM biotin. Each biotin depletion strategy was tested once.

## 'FAMA interactome' and 'nuclear proteome' PL experiments

### Seedling growth and biotin treatment

Approximately 120 µl of seeds per sample and line were surface sterilized with ethanol and bleach and stratified for 2 days. The seeds were then spread on filter paper (Whatman Shark Skin Filter Paper, GE Healthcare 10347509) that was placed on ½ MS plates containing 0.5% sucrose and grown in a growth chamber under long-day conditions for 5 days. For biotin treatment, seedlings were carefully removed from the filter paper, transferred into beakers, covered with 40 ml of a 50 µM biotin solution and incubated in the growth chamber for 0.5 or 3 hr. The biotin solution was removed and seedlings were quickly rinsed with ice cold water and washed tree times for 3 min in approximately 200 ml of ice cold water to stop the labeling reaction (*Branon et al., 2018*) and to remove excess biotin. The seedlings were then dried with paper towels, a sample was taken for immunoblots and the remaining seedlings were split into aliquots of about 1.5 g fresh weight for the three biological replicates, frozen in liquid nitrogen, ground to a fine powder and stored at −80°C until further use. Untreated 0 h samples were frozen directly after harvest. Treatment was timed to keep harvesting times as close together as manageable. Small samples for immunoblots were taken after treatment. The number of replicates was chosen to provide statistical power but to keep the experimental cost and sample handling in a feasible range.

### Affinity purification of biotinylated proteins

For the affinity purification with streptavidin beads, 3 ml of densely packed ground plant material were resuspended in 2 ml extraction buffer (50 mM Tris pH 7.5, 150 mM NaCl, 0.1% SDS, 1% Triton-X-100, 0.5% Na-deoxycholate, 1 mM EGTA, 1 mM DTT, 1x complete, 1 mM PMSF) and incubated on a rotor wheel at 4°C for 10 min. 1 µl Lysonase (Millipore) was added to digest cell walls and DNA/RNA and the suspension was incubated on the rotor wheel at 4°C for another 15 min. The extracts were then distributed into 1.5 ml reaction tubes and sonicated in an ice bath four times for 30 s on high setting using a Bioruptor UCD-200 (Diagenode) with 1.5 min breaks on ice. The suspension was centrifuged for 15 min at 4°C and 15,000 g to remove cell debris and the clear supernatant was applied to a PD-10 desalting column (GE healthcare) to remove excess free biotin using the gravity protocol according to the manufacturer's instructions. Briefly, the column was equilibrated with five volumes of ice cold equilibration buffer (extraction buffer without complete and PMSF), 2.5 ml of the protein extract were loaded and proteins were eluted with 3.5 ml equilibration buffer. The protein

concentration of the protein extract was then measured by Bradford (BioRad protein assay). The protein extract was diluted 1:5 to avoid interference of buffer components with the Bradford assay. A volume of each protein extract corresponding to 16 mg protein was transferred into a new 5 ml LoBind tube (Eppendorf) containing Dynabeads MyOne Streptavidin C1 (Invitrogen) from 200 µl bead slurry that were pre-washed with extraction buffer. Complete protease inhibitor and PMSF were added to reach final concentrations of 1x complete and 1 mM PMSF, respectively, and the samples were incubated on a rotor wheel at 4°C overnight (16 h). The next day, the beads were separated from the protein extract on a magnetic rack and washed as described in *Branon et al. (2018)* with 1 ml each of the following solutions by incubating on the rotor wheel for 8 min and removing the wash solution: 2x with cold extraction buffer (beads were transferred into a new tube the first time), 1x with cold 1 M KCl, 1x with cold 100 mM $Na_2CO_3$, 1x with 2M Urea in 10 mM Tris pH 8 at room temperature and 2x with cold extraction buffer without complete and PMSF. 2% of the beads were boiled in 50 µl 4x Laemmli buffer supplemented with 20 mM DTT and 2 mM biotin at 95°C for 5 min for immunoblots. The rest of the beads was spun down to remove the remaining wash buffer and stored at −80°C until further processing. Sample prep was done in three batches with one replicate of each sample in each batch. Samples from replicate 2 (batch 2) were used for immunoblots to confirm the success of the procedure.

## MS sample preparation

For on-beads tryptic digest, the frozen streptavidin beads were thawed and washed twice each with 1 ml 50 mM Tris pH 7.5 (transferred to new tube with first wash) and 1 ml 2 M Urea in 50 mM Tris pH 7.5. The buffer was removed and replaced by 80 µl Trypsin buffer (50 mM Tris pH 7.5, 1 M Urea, 1 m M DTT, 0.4 µg Trypsin). The beads were then incubated for 3 h at 25°C with shaking, and the supernatant was transferred into a fresh tube. The beads were washed twice with 60 µl 1 M Urea in 50 mM Tris pH 7.5 and all supernatants were combined (final volume 200 µl). The combined eluates were first reduced by adding DTT to a final concentration of 4 mM and incubating at 25°C for 30 min with shaking and then alkylated by adding Iodoacetamide to a final concentration of 10 mM and incubating at 25°C for 45 min with shaking. Finally, another 0.5 µg Trypsin were added and the digest completed by overnight (14.5 h) incubation at 25°C with shaking. The digest was acidified by adding formic acid to a final concentration of ~ 1% and desalted using OMIX C18 pipette tips (10–100 µL, Agilent). C18 desalting tips were first activated by twice aspiring and discarding 200 µl buffer B2 (0.1% formic acid, 50% acetonitrile) and equilibrated by four times aspiring and discarding 200 µl buffer A2 (0.1% formic acid). Peptides were bound by aspiring and dispensing the sample eight times. Then, the tip was washed by 10 times aspiring and discarding 200 µl buffer A2 and the peptides were eluted by aspiring and dispensing 200 µl buffer B2 in a new tube for eight times. The desalted peptides were dried in a speed vac and stored at −80°C until further processing.

## LC-MS/MS

For LC-MS/MS analysis, peptides were resuspended in 0.1% formic acid. Samples were analyzed on a Q-Exactive HF hybrid quadrupole-Orbitrap mass spectrometer (Thermo Fisher), equipped with an Easy LC 1200 UPLC liquid chromatography system (Thermo Fisher). Peptides were separated using analytical column ES803 (Thermo Fisher). The flow rate was 300 nl/min and a 120 min gradient was used. Peptides were eluted by a gradient from 3% to 28% solvent B (80% acetonitrile, 0.1% formic acid) over 100 min and from 28% to 44% solvent B over 20 min, followed by short wash at 90% solvent B. Precursor scan was from mass-to-charge ratio (m/z) 375 to 1600 and top 20 most intense multiply charged precursors were selection for fragmentation. Peptides were fragmented with higher energy collision dissociation (HCD) with normalized collision energy (NCE) 27.

## MS data analysis – protein identification and label-free quantification

Protein identification and label-free quantification (LFQ) were done in MaxQuant (version 1.6.2.6) (*Tyanova et al., 2016a*) using default settings with minor modifications. The datasets for the 'FAMA interactome' and 'nuclear proteome' experiments were searched separately. For the 'FAMA interactome' PL experiment, all samples were sufficiently similar (majority of proteins expected to be equally abundant across samples). Therefore (global) data normalization was done in MaxQuant as part of the LFQ algorithm (*Cox et al., 2014*). For the 'nuclear proteome' PL experiment, in which

the $_{UBQ}$nucTbID samples were markedly different from the other samples (majority of proteins cannot be assumed to be equally abundant across samples), normalization was skipped during LFQ and done separately as described later in the data analysis section. MaxQuant settings for LFQ were as follows. Methionine oxidation and N-terminal acetylation were set as variable modifications and Carbamidomethylcysteine as fixed modification. Maximum number of modifications per peptide was five. Trypsin/P with a maximum of two missed cleavages was set as digest. Peptides were searched against the latest TAIR10 protein database containing a total of 35,386 entries (TAIR10_pep_20101214, updated 2011-08-23, www.arabidopsis.org) plus a list of likely contaminants containing Trypsin, human Keratin, streptavidin, YFP, TbID-mVenus and TbID-YFP and against the contaminants list of MaxQuant. Minimum allowed peptide length was seven. FTMS and TOF MS/MS tolerance were 20 ppm and 40 ppm, respectively, and the peptide FDR and protein FDR were 0.01. Unique and razor peptides were used for protein quantification. LFQ minimum ratio count was set to 2 and fast LFQ was active with a minimum and average number of neighbors of 3 and 6, respectively. For the 'nuclear proteome' experiment 'skip normalization' was enabled to produce non-normalized LFQ values. Match between runs and second peptides were checked. The mass spectrometry proteomics data (raw data, MaxQuant analysis, normalization results (only 'nuclear proteome' experiment) and the following data analysis in Perseus) have been deposited to the ProteomeXchange Consortium (http://proteomecentral.proteomexchange.org) via the PRIDE partner repository (*Vizcaíno et al., 2013*) with the dataset identifiers PXD015161 ('FAMA interactome' experiment) and PXD015162 ('nuclear proteome' experiment).

## MS data analysis – identification of enriched proteins in the 'FAMA interactome' experiment

Filtering and statistical analysis were done with Perseus (version 1.6.2.3) (*Tyanova et al., 2016a*). The 'proteinGroups.txt' output file from MaxQuant was imported into Perseus using the LFQ intensities as Main category. The data matrix was filtered to remove proteins marked as 'only identified by site', 'reverse' and 'potential contaminant'. LFQ values were log2 transformed and proteins that were not identified/quantified in all three replicates of at least one time point of one genotype (low confidence) were removed. The clustering analysis shown in *Figure 5—figure supplement 2* and the multi scatterplot shown in *Figure 5—figure supplement 3*, were done at this point, using the built-in 'Hierarchical clustering' function with the standard settings (Euclidean distance with average linkage, no constraints and preprocessing of k-means) and the 'Multi scatter plot' function (equal ranges, Pearson correlation values displayed), respectively. Next, missing values were imputed for statistical analysis using the 'Replace missing values from normal distribution' function (settings: width = 0.3, down shift = 1.8, mode = total matrix) and principal component analysis was done.

### Protein enrichment in FAMA-TbID samples

To identify proteins enriched in TbID-expressing samples versus the WT control and to remove proteins that bind the beads non-specifically, unpaired two-sided Students t-tests were performed comparing the FAMA-TbID or $_{FAMA}$nucTbID with the corresponding WT samples at each time point. An FDR of 0.05 (integrated modified permutation-based FDR ('Significance Analysis of Microarrays' (SAM) method)) and an S0 of 0.5 with 250 randomizations were chosen as cutoff for multiple sample correction. As an additional filter, only proteins that were quantified in all three replicates of the FAMA-TbID or $_{FAMA}$nucTbID samples at this time point were used for the tests (identified with high confidence; see *Supplementary file 2* – Table 2 for number of proteins used in each test). Significantly enriched proteins were used for hierarchical clustering with standard settings, using Z-transformed non-imputed LFQ values. To further identify proteins that are significantly enriched in FAMA-TbID versus $_{FAMA}$nucTbID and to remove proteins that are stochastically labeled by FAMA-TbID, corresponding t-tests were performed on the reduced dataset using the same parameters as for the first t-tests. Finally, to filter out proteins that were not enriched in at least one of the two treatments compared to untreated FAMA-TbID, unpaired two-sided Students t-tests were performed on the remaining proteins comparing the samples at the 0.5 and 3 h time points to the 0 h time point using a p-value of 0.5 as cutoff.

### Protein enrichment over time

To identify proteins enriched after short or long biotin treatment, unpaired two-sided Students t-tests were performed comparing the 0.5 and 3 h treatment samples of WT, FAMA-TbID and $_{FAMA}$-nucTbID to the corresponding 0 h samples. An FDR of 0.05 and an S0 of 0.5 with 250 randomizations were chosen as cutoff for multiple sample correction. Only proteins that were quantified in all three replicates of at least one of the two groups were tested (see *Supplementary file 2* – Table 3 for number of proteins used in each test).

### Plots and heatmaps

Scatterplots and PCA plots were made in RStudio (version 1.1.463, *RStudio Team, 2016*, R version 3.5.1, *R Development Core Team, 2018*) using t-test and PCA results exported from Perseus. Hierarchical clustering was done in Perseus and the plots were exported as pdf.

## MS data analysis – data normalization and identification of enriched proteins in the 'nuclear proteome' experiment

Since the samples of the 'nuclear proteome' experiment were quite different in the number and abundance of identified proteins (majority of proteins cannot be assumed to be equally abundant across samples) and each sample was measured as one fraction, normalization was skipped in Max-Quant and done after peptide identification and quantification using the non-normalized LFQ values. Search for a suitable normalization method and normalization were done with Normalyzer (version 1.1.1) (*Chawade et al., 2014*), a tool to help select an optimal normalization method for a given dataset from a set of commonly used global and local normalization methods. Subsequent data filtering and statistical analysis were done with Perseus (version 1.6.2.3) (*Tyanova et al., 2016b*). First, the 'proteinGroups.txt' output file from MaxQuant was imported into Perseus using the LFQ intensities as Main category. The data matrix was filtered to remove proteins marked as 'only identified by site', 'reverse' and 'potential contaminant' (reduction of the dataset from 5038 proteins to 4964 proteins) and the LFQ values of all remaining proteins were exported for normalization with Normalyzer (http://normalyzer.immunoprot.lth.se/normalize.php). Based on the assumption that the majority of proteins in our dataset is differentially abundant and on the performance of the supported normalization methods, we determined that LOESS-R (local normalization) is most suitable for the dataset. LOESS-R normalized LFQ values were then imported back into Perseus and log2 transformed. Proteins that were not identified/quantified in all three replicates of at least one genotype (low confidence) were removed. The clustering analysis shown in *Figure 6—figure supplement 1*, as well as the multi scatterplot shown in *Figure 6—figure supplement 2*, were done at this point, using the built-in 'Hierarchical clustering' and 'Multi scatter plot' functions as described for the 'FAMA interactome' experiment. Next, missing values were imputed for statistical analysis using the 'Replace missing values from normal distribution' function (settings: width = 0.3, down shift = 1.8, mode = total matrix) and principal component analysis was done.

### Protein enrichment in nuclear TbID samples

To identify proteins enriched in nuclear TbID-expressing samples versus the WT control, unpaired two-sided Students t-tests were performed comparing the $_{UBQ}$nucTbID or $_{FAMA}$nucTbID with WT samples. An FDR of 0.05 (integrated modified permutation-based FDR ('Significance Analysis of Microarrays' (SAM) method)) and an S0 of 0.5 with 250 randomizations were chosen as cutoff for multiple sample correction. As an additional filter, only proteins that were quantified in all three replicates of the $_{UBQ}$nucTbID or $_{FAMA}$nucTbID sample were used for the tests (identified with high confidence; see *Supplementary file 3* – Tables 2 and 3 for number of proteins used in each test). To further identify proteins that are specific for or highly enriched in FAMA nuclei compared to all nuclei, we looked for proteins that were equally or slightly more abundant in the $_{FAMA}$nucTbID than the $_{UBQ}$nucTbID samples, by performing unpaired two-sided t-tests between $_{FAMA}$nucTbID and $_{UBQ}$-nucTbID on the reduced dataset. As cutoff, an FDR of 0.01 and S0 of 0.1 were chosen. Protein not significantly enriched in the $_{UBQ}$nucTbID samples were used for hierarchical clustering with standard settings, using Z-transformed non-imputed LFQ values.

## Plots and heatmaps
Scatterplots and PCA plots were made as described for the 'FAMA interactome' experiment.

## Classification of the subcellular localization of nuclear proteins form PL
Proteins identified by PL with nuclear TbID were divided into three broad categories: (1) previously identified in the nucleus experimentally, (2) not found experimentally but predicted to be in the nucleus and (3) predicted to be elsewhere. To determine proteins in the first category, a list of proteins identified in published nuclear- or subnuclear proteomics studies and of proteins detected in the nucleus in localization studies with fluorescent proteins (from SUBA4; *Hooper et al., 2017*; http://suba.live/; date of retrieval: February 5 2019) was compiled (see *Supplementary file 3* – Table 4). Localization predictions for proteins not identified experimentally in the nucleus was done using the SUBAcon prediction algorithm (*Hooper et al., 2014*) on SUBA4 (date of retrieval: February 5 2019). SUBAcon provides a consensus localization based on 22 different prediction algorithms and available experimental data.

## Gene ontology analysis
Proteins that were significantly enriched in $_{FAMA}$nucTbID and $_{UBQ}$nucTbID compared to WT (FAMA- and global nuclear proteins) were used for GO analysis with AgriGO v2 (*Tian et al., 2017*) with the following settings: Singular enrichment analysis (SAE) with TAIR10_2017 as background, statistical method: fisher, multi-test adjustment: Yekutieli (FDR), significance level: 0.05, minimum number of entries: five. Significantly enriched GO terms from AgriGO v2 (see *Supplementary file 3* – Table 7) were visualized using REViGO (*Supek et al., 2011*) with the following settings: list size = medium, associated numbers = p-values, database with GO term size = *Arabidopsis thaliana*, semantic similarity measure = SimRel. The R script provided by REViGO was used to draw the TreeMap plot in RStudio. Very small labels were removed and label size was adjusted in Adobe Illustrator.

## Evaluation of selected FAMA interaction partner candidates by Y2H and PL assays in *N. benthamiana*
### Y2H assays
FAMA in pXDGATcy86 (Gal4 DNA binding domain) was tested for interaction with ICE1, MUTE, SEU, LUH, BZR1 and BIM1 in pGADT7-GW (Gal4 activation domain). Transformation of yeast strain AH109 with bait and prey plasmids (empty or containing one of the listed genes), selection for transformants and testing of pair-wise interactions by growth complementation assays on nutrient selective media and with X-α-Gal was done as described in the Matchmaker GAL4 Two-Hybrid System three manual (Clontech). To compensate for autoactivation of BD-FAMA, 3-AT was added to selective plates at different concentrations for growth complementation assays.

### PL assays in *N. benthamiana*
*N. benthamiana* ecotype NB-1 was co-transformed with three or four constructs as described before by infiltrating young leaves with suspensions of Agrobacteria carrying the following vectors: (1) one of UBQ10pro::cFAMA-TbID-Venus in pK7m34GW,0 or UBQpro::TbID-YFP$_{NLS}$ in R4pGWB601, (2) one or two of UBQ10pro::ICE1-3xHA, UBQ10pro::MUTE-3xHA, UBQ10pro::LUH-3xHA in R4pGWB613 and UBQ10pro::SEU-4xmyc in R4pGWB616 (Agrobacterium strain GV3101) and (3) an Agrobacterium strain carrying a 35S::p19 plasmid (tomato bushy stunt virus (TBSV) protein p19) for protein stabilization. Two days after infiltration, expression was confirmed by epifluorescence microscopy and 5 mm wide leaf discs were harvested and flash frozen. 2 × 15 leaf discs from two different leaves were used per sample for the pulldown experiments. Leaf discs were ground to a fine powder in a ball mill. Proteins were extracted by adding 600 µl ice cold extraction buffer (50 mM Tris pH 7.5, 150 mM NaCl, 0.1% SDS, 0.5% sodium deoxycholate, 1% Triton-X-100, 1 mM EDTA, 1 mM EGTA, 1 mM DTT, 1 mM PMSF, 1x complete protease inhibitor). Extracts were sonicated in an ice bath four times for 30 s on high setting using a Bioruptor UCD-200 (Diagenode) with 1.5 min breaks. Suspensions were centrifuged twice for 10 min at 4°C and 15,000 g to remove cell debris. The cleared supernatants were mixed with pre-washed Dynabeads MyOne Streptavidin C1 (Invitrogen) from 20 µl bead slurry and incubated on a rotor at 4°C for 2 h. Beads were washed tree times with 300 µl cold extraction buffer, once with 2 M Urea in 10 mM Tris pH 8 (room temperature) and once more

with cold extraction buffer. Proteins were eluted by boiling the beads in 4x Leammli buffer supplemented with 20 mM DTT and 2 mM biotin for 10 min. Protein abundance in the extracts and bead elutions were tested by immune blotting.

## Microscopy

Brightfield and epifluorescence images of *N. benthamiana* leaves and Arabidopsis seedlings, leaves and flowers were taken with a Leica DM6B microscope using a Leica CRT6 LED light source. Confocal microscopy images of Arabidopsis seedlings expressing different biotin ligase constructs were taken with a Leica SP5 microscope. For confocal microscopy, cell walls were stained with propidium iodide (Molecular Probes) by incubating in a 0.1 mg/ml solution for three to five minutes. Images were processed in FIJI (ImageJ) (*Schindelin et al., 2012*). Several independent Arabidopsis lines and transiently transformed *N. benthamiana* leaves were analyzed. Images shown in figures and figure supplements are representative.

## Immunoblots

Samples for immunoblots were prepared by resuspending frozen and ground plant material from biotin treatment assays with 1x Leammli buffer (60 mM Tris pH 6.8, 2% SDS, 10% glycerol, 2.5% beta-mercaptoethanol, 0.025% bromphenol blue) or mixing protein extracts 1:1 with 2x Leammli buffer and boiling the samples for five minutes at 95°C. Proteins bound to SA or GFP-Trap beads were eluted from the beads as described in the respective Materials and methods sections. Proteins were separated by SDS-PAGE and blotted onto Immobilon-P PVDF membrane (0.45 µm, Millipore) using a Trans-Blot Semi-Dry transfer Cell (BioRad). The following antibodies were used: Streptavidin-HRP (S911, Thermo Fisher Scientific), Rat monoclonal anti-GFP antibody (3H9, Chromotek), Rat Anti-HA High Affinity (11867423001, Roche), Myc-Tag (71D10) Rabbit mAb (2278 S Cell Signaling), AffiniPure Donkey Anti-Rat IgG-HRP (712-035-153, Jackson Immuno Research Laboratories), Rabbit Anti-Rat IgG-HRP (A5795, Sigma), Goat anti-Rabbit IgG-HRP (AS09 602, Agrisera). Blots were probed with primary antibodies overnight at 4°C or for up to 1 h at room temperature and with the secondary antibody for 1 h at room temperature and incubated with ECL Western blotting substrates according to the manufacturer's instructions. Signals were detected on X-ray films or on a ChemiDoc MP Imaging System (BioRad).

## FAMA-CFP AP-MS experiments

### Seedling growth and crosslinking

For the two FAMA-CFP AP-MS experiments, FAMApro::FAMA-CFP complementing the homozygous *fama-1* mutant and two lines expressing nuclear GFP under an early and late stomatal lineage specific promoter (SPCHpro::GFP$_{NLS}$ and MUTEpro::GFP$_{NLS}$) were used. Large amounts of seedlings were grown for four days in liquid culture in ½ MS + 0.5% sucrose (AP-MS experiment 1) or on filter paper that was placed on ½ MS + 0.5% sucrose plates (AP-MS experiment 2). Seedlings were then treated with the proteasome inhibitor MG-132 by submerging them in liquid ½ MS + 0.5% sucrose and 10 µM MG-132 for 2.5 (AP-MS 1) or 2 (AP-MS 2) hours to increase the abundance of FAMA and other unstable proteins and crosslinked with formaldehyde. Crosslinking was done by removing excess liquid, submerging the seedlings in crosslinking buffer (25 mM Hepes pH 7.5, 0.5 mM EDTA, 1 mM PMSF, 10 mM MgCl$_2$, 75 mM NaCl) containing 0.25% (AP-MS 1) or 0.125% (AP-MS 2) formaldehyde and vacuum infiltrating for 15 min on ice. Crosslinking was stopped by adding glycine to a final concentration of 120 mM and vacuum infiltrating for another 10 min. Seedlings were washed with 800 ml ice cold H$_2$O and frozen in liquid nitrogen. Seedlings were treated in two batches per experiment.

### Affinity purification

For AP, the frozen plant material was ground to a fine powder. Two biological replicates per genotype were used for AP-MS 1 and four replicates of the FAMA-CFP line and two of each of the control lines were used for AP-MS 2. AP-MS 1 was done to test overall performance of the experimental setup before doing a larger scale experiment. Therefore, only a small number of replicates was used. To increase statistical power in AP-MS 2, the replicate number was doubled. Approximately 15 g plant material (from one or two of the treatment batches) per sample were resuspended in 25

ml extraction buffer (25 mM Tris pH 7.5, 10 mM MgCl$_2$, 0.5 mM EGTA, 75 mM NaCl, 0.5/1% Triton-X-100, 1 mM NaF, 0.5 mM Na$_3$VO$_4$, 15 mM beta glycerophosphate, 1 mM DTT, 1 mM PMSF, 1x complete proteasome inhibitor) and incubated on a rotor wheel at 4°C for 30 to 45 min to thaw. The extracts were sonicated in 15 ml tubes in an ice bath using Bioruptor UCD-200 (Diagenode) on high setting. For experiment 1, samples were sonicated twice for five minutes with a 30 s on/off interval and a five-minute break on ice. For experiment 2, the sonication cycles were reduced to 2.5 min with a 30/60 s on/off interval. After sonication, 15 µl Lysonase (Millipore) were added and the extracts were incubated for another 30 min on a rotor wheel at 4°C before centrifuging at 12,000 g and 4°C for 10 min. The supernatants were filtered through two layers of Miracloth (Millipore) and incubated with 25 µl of GFP-Trap_MA beads (ChromoTek), pre-washed with extraction buffer, for five (AP-MS 1) or three (AP-MS 2) hours at 4°C on a rotor wheel. The beads were then washed six times with extraction buffer containing 100 mM NaCl (1 × 15 ml, 5 × 1 ml), resuspended in 50 µl 2x Leammli buffer and boiled for 5 min at 95°C. Samples were stored at −80°C until further processing.

## MS sample preparation

Affinity purified proteins were separated by SDS-PAGE using Mini-PROTEAN TGX gels (BioRad) and stained with the Novex colloidal blue staining kit (Invitrogen). About 1 cm long sample areas were excised from the gel, and used for in-gel Tryptic digestion. Peptides were desalted using C18 Zip-Tips (Millipore). LC-MS/MS was done as described for the PL experiments.

## MS data analysis

Protein identification and label-free quantification (LFQ) were done for both experiments together. RAW files from LC-MS/MS were searched in MaxQuant (version 1.6.2.6) (*Tyanova et al., 2016a*) using the same settings as described for the PL experiments with one difference: LFQ minimum ratio count was set to 1 instead of 2. The mass spectrometry proteomics data (RAW files, MaxQuant search results and Perseus data analysis file) have been deposited to the ProteomeXchange Consortium (http://proteomecentral.proteomexchange.org) via the PRIDE partner repository (*Vizcaíno et al., 2013*) with the dataset identifier PXD015212.

Filtering and statistical analysis were done with Perseus (version 1.6.2.3) (*Tyanova et al., 2016a*). The 'proteinGroups.txt' output file from MaxQuant was imported into Perseus using the LFQ intensities as Main category. The data matrix was filtered to remove proteins marked as 'only identified by site', 'reverse' and 'potential contaminant'. LFQ values were log2 transformed and samples from AP-MS 1 and AP-MS 2 were separated for individual analysis. Proteins that were not identified/quantified in at least two (AP-MS 1) or three (AP-MS 2) FAMA-CFP or control samples (low confidence) were removed. Missing values were imputed for statistical analysis using the 'Replace missing values from normal distribution' function with the following settings: width = 0.3, down shift = 1.8 and mode = total matrix. Significantly enriched proteins were identified by unpaired two-sided Students t-tests comparing the FAMA-CFP with the control samples and using the integrated modified permutation-based FDR with 250 randomizations for multiple sample correction. Only proteins that were identified in two (AP-MS 1) or at least three (AP-MS 2) replicates of the FAMA-CFP samples were used for the test. FDR/S0 combinations were chosen to get a minimum number of 'false negatives' (statistically enriched proteins in controls). For AP-MS 1, the FDR was 0.2 and S0 was 0.5 and for AP-MS 2, the FDR was 0.01 and S0 was 0.5. Scatterplots with log2-fold changes and -log10 p-values form t-tests were done in RStudio.

## Primers

| Primer name | Purpose | Sequence |
| --- | --- | --- |
| BirA-fw | Cloning | CAGGCGCGCCGGTGGAGGCGGTTCAGGAGGTGGCATGGGCAAGCCCATCCCCAAC |
| BirA-NES-rv | Cloning | CTGGCGCGCCCACCTCCGCCGCTTCCACCGCCTCCGTCCAGGGTCAGGCGCTCCAG |

*Continued on next page*

*Continued*

| Primer name | Purpose | Sequence |
| --- | --- | --- |
| BirA-rv | Cloning | TGGCGCGCCCACCTCCGC CGCTTCCACCGCCTCCCTT TTCGGCAGACCGCAGACTGATTT |
| pDONR-mut-AscI-fw | Cloning | GTACAAAGTGGCTGGGC GCGCCTCCATGGTGAGCAAGG |
| pDONR-mut-AscI-rv | Cloning | CCAGCCACTTTGTACA AGAAAGTTGAACGAG |
| gFAMA-fw | Cloning | CACCATGGATAAAGAT TACTCGGTACGTACG |
| gFAMA-rv | Cloning | AGTAAACACAATATTT CCCAGGTTAGAGC |
| BirA-YFPnls-fw | Cloning | GCATGCGGCCGCAT GGGCAAGCCCATC |
| BirA-YFPnls-rv | Cloning | GATACCATGGAACC TCCGCCGCTTCC |
| MUTE-fw | Cloning | CACCATGTCTCACAT CGCTGTTGAAAGGAATCG |
| MUTE-rv | Cloning | ATTGGTAGAGACGA TCACTTCATCAGAC |
| ICE1-fw | Cloning | CACCATGGGTCTT GACGGAAACAATGG |
| ICE1-rv | Cloning | GATCATACCAGCA TACCCTGCT |
| SEU-fw | Cloning | CACCATGGTACCA TCAGAGCCGCC |
| SEU-rv | Cloning | CGCGTTCCAATCAAA ATTGTTGAAAC |
| LUH-fw | Cloning | CACCATGGCTCAGA GTAATTGGGAAG |
| LUH-rv | Cloning | CTTCCAAATCTTTAC GGATTTGTCATG |
| BZR1-fw | Cloning | CACCATGACTTCGGA TGGAGCTACG |
| BZR1-rv | Cloning | ACCACGAGCCTT CCCATTTC |
| BIM1-fw | Cloning | CACCATGGAGCT TCCTCAACCTCGTC |
| BIM1-rv | Cloning | CTGTCCCGTCTT GAGCCGTT |
| *fama1*-RP | Genotyping | CAATACAAAAA GCTCCCCTCAC |
| *fama1*-LBb1.3 | Genotyping | ATTTTGCCGA TTTCGGAAC |

# ʹPL toolboxʹ vectors generated for common use

**Entry vectors for creating protein of interest-TbID/mTb + YFP/mVenus fusions**

| Plasmid name | Gateway sites | Application | Addgene ID |
| --- | --- | --- | --- |
| pENTR_L1-YFP-Turbo-NES-L2 | attL1-attL2 | For N-terminal YFP-TbID fusion to non-nuclear proteins | 127349 |

*Continued on next page*

*Continued*

**Entry vectors for creating protein of interest-TbID/mTb + YFP/mVenus fusions**

| | | | |
|---|---|---|---|
| pENTR_L1-YFP-Turbo-L2 | attL1-attL2 | For N-terminal YFP-TbID fusion to nuclear proteins | 127350 |
| pENTR_L1-YFP-miniTurbo-NES-L2 | attL1-attL2 | For N-terminal YFP-mTb fusion to non-nuclear proteins | 127351 |
| pENTR_L1-YFP-miniTurbo-L2 | attL1-attL2 | For N-terminal YFP-mTb fusion to nuclear proteins | 127352 |
| pDONR_P2R-P3_R2-Turbo-NES-mVenus-STOP-L3 | attR2-attL3 | For C-terminal TbID-mVenus fusion to non-nuclear proteins | 127353 |
| pDONR_P2R-P3_R2-Turbo-mVenus-STOP-L3 | attR2-attL3 | For C-terminal TbID-mVenus fusion to nuclear proteins | 127354 |
| pDONR_P2R-P3_R2-miniTurbo-NES-mVenus-STOP-L3 | attR2-attL3 | For C-terminal mTb-mVenus fusion to non-nuclear proteins | 127355 |
| pDONR_P2R-P3_R2-miniTurbo-mVenus-STOP-L3 | attR2-attL3 | For C-terminal mTb-mVenus fusion to nuclear proteins | 127356 |

**Entry vectors for expressing BirA*/TbID/mTb + YFP under a promoter of choice**

| Plasmid name | Gateway sites | application | Addgene ID |
|---|---|---|---|
| pENTR_L1-BirA(R118G)-NES-YFP-STOP-L2 | attL1-attL2 | For expressing cytosolic BirA* under a promoter of choice | 127357 |
| pENTR_L1-Turbo-NES-YFP-STOP-L2 | attL1-attL2 | For expressing cytosolic TbID under a promoter of choice | 127358 |
| pENTR_L1-miniTurbo-NES-YFP-STOP-L2 | attL1-attL2 | For expressing cytosolic mTb under a promoter of choice | 127359 |
| pENTR_L1-BirA(R118G)-YFP-NLS-STOP-L2 | attL1-attL2 | For expressing nuclear BirA* under a promoter of choice | 127360 |
| pENTR_L1-Turbo-YFP-NLS-STOP-L2 | attL1-attL2 | For expressing nuclear TbID under a promoter of choice | 127361 |
| pENTR_L1-miniTurbo-YFP-NLS-STOP-L2 | attL1-attL2 | For expressing nuclear mTb under a promoter of choice | 127362 |

**Binary plant transformation vectors for ubiquitous expression of BirA*/TbID/mTb + YFP/mVenus**

| Plasmid name | Resistance in plants | application | Addgene ID |
|---|---|---|---|
| R4pGWB601_UBQ10p-BirA(R118G)-NES-YFP | BASTA | For expressing cytosolic BirA* under the UBQ10 promoter | 127363 |
| R4pGWB601_UBQ10p-BirA(R118G)-YFP-NLS | BASTA | For expressing nuclear BirA* under the UBQ10 promoter | 127365 |
| R4pGWB601_UBQ10p-Turbo-NES-YFP | BASTA | For expressing cytosolic TbID under the UBQ10 promoter | 127366 |
| R4pGWB601_UBQ10p-Turbo-YFP-NLS | BASTA | For expressing nuclear TbID under the UBQ10 promoter | 127368 |
| R4pGWB601_UBQ10p-miniTurbo-NES-YFP | BASTA | For expressing cytosolic mTb under the UBQ10 promoter | 127369 |
| R4pGWB601_UBQ10p-miniTurbo-YFP-NLS | BASTA | For expressing nuclear mTb under the UBQ10 promoter | 127370 |

# Acknowledgements

We thank Annika Weimer for suggestions for proximity labeling construct design and testing of biotin ligase activity and other previous and current members of the Bergmann lab for creating tools that enabled this work and for critical discussion.

## Additional information

### Competing interests
Dominique C Bergmann: Reviewing editor, *eLife*. The other authors declare that no competing interests exist.

### Funding

| Funder | Grant reference number | Author |
| --- | --- | --- |
| Howard Hughes Medical Institute | | Dominique C Bergmann |
| Austrian Science Fund | J4019-B29 | Andrea Mair |
| National Institutes of Health | RO1-CA186568 | Alice Y Ting |
| Carnegie Institution of Washington | | Shou-ling Xu |
| Massachusetts Institute of Technology | Lester Wolfe Fellowship | Tess C Branon |
| Massachusetts Institute of Technology | Dow Graduate Research Fellowship | Tess C Branon |

The funders had no role in study design, data collection and interpretation, or the decision to submit the work for publication.

### Author contributions
Andrea Mair, Conceptualization, Data curation, Formal analysis, Funding acquisition, Validation, Visualization, Methodology, Writing—original draft, Project administration, Writing—review and editing; Shou-Ling Xu, Data curation, Formal analysis, Methodology, Writing—original draft, Writing—review and editing; Tess C Branon, Alice Y Ting, Resources, Writing—review and editing; Dominique C Bergmann, Conceptualization, Resources, Funding acquisition, Writing—original draft, Project administration, Writing—review and editing

### Author ORCIDs
Andrea Mair 
Shou-Ling Xu 
Alice Y Ting 
Dominique C Bergmann 

### Decision letter and Author response
Decision letter https://doi.org/10.7554/eLife.47864.sa1
Author response https://doi.org/10.7554/eLife.47864.sa2

## Additional files

### Supplementary files
• Supplementary file 1. Non-cropped immunoblots.

• Supplementary file 2. XLS file containing the following additional protein lists for the 'FAMA interactome' experiment (*Figure 5* and supplements). Table 1, 'identified proteins': Proteins identified in the 'FAMA interactome' experiment; Table 2, 'FAMA-TbID enriched': Proteins significantly enriched in FAMA-TbID vs. WT and $_{FAMA}$nucTbID samples and in $_{FAMA}$nucTbID vs. WT; Table 3, 'enrichment over time': Proteins significantly enriched in WT, FAMA-TbID, and $_{FAMA}$nucTbID after biotin treatment compared to the no-biotin control; Table 4, 'FAMA complex candidates': FAMA interaction candidates from *Figure 5* and Table 1; Table 5, 'FAMA AP-MS': Proteins significantly enriched in AP-MS experiments with FAMA-CFP

- Supplementary file 3. XLS file containing the following additional protein lists for the 'nuclear proteome' experiment (*Figure 6* and supplements). Table 1, 'identified proteins': Proteins identified in the 'nuclear proteome' experiment; Table 2, 'enriched with $_{UBQ}$nucTbID': Proteins significantly enriched in $_{UBQ}$nucTbID vs. WT samples; Table 3, 'enriched with $_{FAMA}$nucTbID': Proteins significantly enriched in $_{FAMA}$nucTbID vs. WT and $_{FAMA}$nucTbID vs. $_{UBQ}$nucTbID samples; Table 4, 'published nuclear proteomes': Proteins found in proteomics studies of purified nuclei or sub-nuclear compartments by mass spectrometry or in localization studies with fluorescent protein fusions; Table 5, 'localization prediction': Protein localization data used for pie charts in *Figure 6*; Table 6, 'published GC proteome': Proteins found in published guard cell proteomics experiments; Table 7, 'GO term enrichment': Enriched GO terms of nuclear proteins in *Figure 6* and supplements; Table 8, 'nuclear compartments': Selected marker proteins and protein classes for different nuclear compartments and domains identified in the $_{UBQ}$nucTbID and $_{FAMA}$nucTbID dataset

- Transparent reporting form

### Data availability

MS data have been deposited to the ProteomeXchange Consortium (http://proteomecentral.proteomexchange.org) via the PRIDE partner repository (Vizcaino et al. 2013).

The following datasets were generated:

| Author(s) | Year | Dataset title | Dataset URL | Database and Identifier |
|---|---|---|---|---|
| Andrea Mair, Shouling Xu, Dominique C Bergmann | 2019 | Proximity labeling dataset for the 'FAMA interactome' experiment | http://proteomecentral.proteomexchange.org/cgi/GetDataset?ID=PXD015161 | ProteomeXchange, PXD015161 |
| Andrea Mair, Shouling Xu, Dominique C Bergmann | 2019 | Proximity labeling dataset for the 'nuclear proteome' experiment | http://proteomecentral.proteomexchange.org/cgi/GetDataset?ID=PXD015162 | ProteomeXchenge, PXD015162 |
| Andrea Mair, Shouling Xu, Dominique C Bergmann | 2019 | FAMA-CFP AP-MS datasets | http://proteomecentral.proteomexchange.org/cgi/GetDataset?ID=PXD015212 | ProteomeXchange Consortium, PXD015212 |

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
