## [Decision Letter]

Thank you for submitting your article "Proximity labeling of protein complexes and cell type-specific organellar proteomes in Arabidopsis enabled by TurboID" for consideration by *eLife*. Your article has been reviewed by three peer reviewers, one of whom is a member of our Board of Reviewing Editors, and the evaluation has been overseen by Christian Hardtke as the Senior Editor. The reviewers have opted to remain anonymous.

The reviewers have discussed the reviews with one another and the Reviewing Editor has drafted this decision to help you prepare a revised submission.

Summary:

The manuscript by Mair et al., (Tools and Resources) has undertaken a robust assessment of an improved proximity labeling with biotin (BioID) approach in plants. Few studies in plants have employed BioID, and here the authors have presented a laborious collection of data outlining conditions necessary for the implementation of these improved methods (TurboID and miniTurboID) to identify new protein-protein interactions in plants in both stable and transient backgrounds. The authors start out by comparing cytosolic and nuclear localized BirA*- and TurboID- and MiniTurboID-YFP fusion expressed transiently in *N. benthamiana* and stably expressed in Arabidopsis transgenic lines. Using carefully designed experiments variables such as labelling temperature, biotin concentration, labelling time are tested and optimal conditions identified for TurboID as well as miniTurbo. This is done in *N. benthamiana* leaves as well as in different developmental stages and organs of Arabidopsis. Optimal conditions for affinity purification of biotinylated proteins and sample processing prior to mass spec analysis are carefully worked out and documented in great detail. Once these conditions are worked out the authors test the TurboID version to characterize a cell type specific transcription factor (FAMA) complex, with a FAMA-TurboID fusion, as well as the nuclear composition of specific guard cells expression transcription factor FAMA (pFAMA:TurboID). This is done in a temporal manor, by using short and medium length labelling times (0.5 and 3 hrs) compared to O hrs as well as comparison to the nuclear proteome (pUBQ:TurboID) and non-transformed negative controls. Both the FAMA complex and the FAMA nucleo-proteome experiments are very well designed and well documented. The FAMA complex experiment after filtering out 'non-specific' biotinylated proteins result in 47 potential FAMA complex associated/neighbourhood proteins.

The manuscript also describes a collection of vectors that have been made available for rapid implementation of their approach of this technique by the community.

Overall, performed experiments were well designed and presented data/information are very valuable for the plant science community to design the TurboID and miniTurbo based PL experiments for their researches.

Essential revisions:

1) Could the authors please explain how normalization of the data during MaxQuant search was done and/or was this done with Perseus? Some samples are very different from control samples, and in these cases application of global normalization across all samples during MaxQuant search (for instance) may bias the results significantly.

2) Please describe the filtering process and methods in more detail. The filtering process (Figure 5 and Figure 6) is difficult to really follow and does rely on numerous factors involved with MS data processing (matching parameters, LFQ parameters). The work has been replicated (n=3) which does add higher levels of certainty in the final results – FAMA interaction candidates and a guard cell nuclear proteome. But the initial datasets are 2511 and 3176 proteins, respectively and are filtered to 47 and 451/1583 proteins. For the GC proteome the authors indicate they found several known nuclear marker for young GCs – in fact they found 3 (subsection “Suitability of TurboID and miniTurboID for application in plants and performance of TurboID in FAMA-complex identification and nuclear proteome analysis”, last paragraph). It would be helpfull in understanding this extensive filtereing if authors could provide the MaxQuant Search files such as 'experimentalDesign', 'parameters' for the reviewers (and include in the Pride data submission).

3) The extensive data filtering to generate a high confidence sets (especially with regard to FAMA interactors) would usually require validation of some of the new candidates. Have the authors conducted any preliminary validations of the new interactors? Even relatively preliminary data would strengthen their claims for this approach. This could be preliminary transient work e.g. co express FAMA-TurboId and Target:Myc, pull down with streptavidin and conduct a western blot with anti-Myc. This would ideally be transient targets rather than complex partners, as this is the articulated advantage of this method over e.g. AP-MS.

4) The access to the raw MS data in PRIDE does not work. Both PRIDE login details resulted in no available data. I would guess data have been updated in the interim and this requires a new username / password to be issued for reviewers. Thus the robustness of the MS data matching employed by the authors could not be evaluated nor examine the quantitative process which is presumably available in a viewable data format.

5) Figure 5—figure supplement 2B: Why is replicate 1 is separated from replicate 2 and 3? It seems that separation by replicates is more significant than separation by genotypes. Is this due to how samples were processed? It was not totally clear from the method. Would be nice to have multi scatter plots as well to show correlation between samples.

6) In my understanding Branon et al. named the new ligases as TurboID(TbID) and miniTurbo(mTb), and therefore authors should use same terms to avoid any confusions.

---

## [Author Response]

Essential revisions:1) Could the authors please explain how normalization of the data during MaxQuant search was done and/or was this done with Perseus? Some samples are very different from control samples, and in these cases application of global normalization across all samples during MaxQuant search (for instance) may bias the results significantly.

Thank you for bringing up this question. We agree that data normalization is a very important step and can have a large impact on the results of any MS study and have therefore carefully considered this issue.

For the MS experiments included in our manuscripts, data normalization was done in MaxQuant as part of the label-free quantification (LFQ) process, which is described in detail in (Cox et al., 2014). We also contacted that study’s author, Jürgen Cox (MPI for Biochemistry, Martinsried) for clarification about normalization. In essence, the MaxLFQ algorithm first normalizes peptide intensities (by pair-wise comparison of peptide intensities between samples and calculation of optimized normalization factors) and then quantifies peptide/protein abundance. It is therefore a global normalization method, as has been pointed out, and requires samples to be overall similar not to introduce a bias. If samples are too different for normalization with MaxLFQ, the algorithm will fail and not terminate successfully (personal communication with Jürgen Cox).

In the case of AP-MS studies, for which MaxQuant is often used, the algorithm uses the fact that pulldown samples usually contain a large body of non-specific proteins (background) that is common to all samples. In our ‘FAMA interactome’ experiment, samples are overall very similar (see heatmaps and multi scatterplots Figure 5—figure supplements 2 and 3), with over 80% of identified proteins found in all of the nine sample groups, including the WT controls, and about 50% being found in all of the 27 samples. Due to the large number of sticky proteins that presumably bound to the beads non-specifically, this dataset meets the criteria that the majority of proteins are expected to be equally abundant in all samples. We therefore believe, that for this dataset normalization in MaxQuant is a valid choice.

Although MaxLFQ normalization did not fail for the ‘nuclear proteome’ dataset, we agree that global normalization is not optimal in this case. We compared non-normalized and normalized data and found that global normalization in MaxQuant leads to a slight overall reduction of LFQ values in _UBQ_nucTbID samples (due to higher protein identifications and quantity) and an underestimation of protein abundance in these samples. We have therefore decided to re-analyze the MS data for this experiment (Figure 6 and Figure 6—figure supplements 1-4). For our new analysis, we used the Normalyzer tool (Chawade, Alexandersson, and Levander, 2014) to determine an optimal normalization method, based on the assumption that the majority of proteins is not equal in all samples and on the performance of the commonly used normalization methods supported by the algorithm (see Materials and methods section subsection “Protein enrichment over time”, for more details). Using LOESS-R (a local normalization method), we see an expected increase of proteins enriched in the _UBQ_nucTbID dataset (632 proteins), and a slight decrease in enriched proteins in the _FAMA_nucTbID dataset (61 proteins lost, 4 proteins gained). FAMA nuclei specific proteins remained largely unchanged, with all three marker proteins included, as did the subcellular distribution of our nuclear proteins based on experimental and prediction data.

Overall, we believe that the new analysis reflects the data better. We would, however, like to point out that despite the shift in numbers of enriched proteins, the major findings and conclusions of the experiment remain unchanged.

To emphasize the importance of data normalization for PL experiments for our readers, we have included a sentence on data normalization in the Discussion (subsection “Considerations for a successful PL experiment”, last paragraph).

2) Please describe the filtering process and methods in more detail. The filtering process (Figure 5 and Figure 6) is difficult to really follow and does rely on numerous factors involved with MS data processing (matching parameters, LFQ parameters). The work has been replicated (n=3) which does add higher levels of certainty in the final results – FAMA interaction candidates and a guard cell nuclear proteome. But the initial datasets are 2511 and 3176 proteins, respectively and are filtered to 47 and 451/1583 proteins. For the GC proteome the authors indicate they found several known nuclear marker for young GCs – in fact they found 3 (subsection “Suitability of TurboID and miniTurboID for application in plants and performance of TurboID in FAMA-complex identification and nuclear proteome analysis”, last paragraph). It would be helpful in understanding this extensive filtereing if authors could provide the MaxQuant Search files such as 'experimentalDesign', 'parameters' for the reviewers (and include in the Pride data submission).

As the reviewers point out, we observe a high number of proteins in our dataset, that were identified across all sample types, including the WT control (see Figure 5—figure supplements 2-3 and Figure 6—figure supplement 1). These likely comprise proteins that bound to the beads non-specifically (sticky proteins and proteins too abundant to be washed off completely) and a small number of naturally biotinylated proteins. These proteins were identified in our first filtering step by comparing TbID samples with WT samples (not enriched in the TbID samples) and removed before further comparisons between TbID samples were made. It is not uncommon for AP-MS experiments that a large fraction of identified proteins binds to the beads, especially when a large amount of beads is used as in our experiments. Although these proteins are useful for MS data normalization, we hope to improve sample purity in future experiments.

To make our filtering process and the aims of each filtering step more accessible to the reader, we have added more detail to the description of the process in the Results section (subsection “Proximity labeling is superior to AP-MS for identification of candidate interactors of FAMA”). The filtering process and the applied statistical tests with cutoff values are also described in the figure legends of Figures 5 and 6 and a detailed description of the whole data analysis (including MS instrument and MS search settings) can be found in the Materials and methods sections.

The MaxQuant search output (containing the ‘parameters’ and ‘summary’ files, as well as all results files) had been included in the PRIDE data submission together with the RAW data files, but were unfortunately inaccessible to reviewers (see response to major point 4). To make it easier to follow our data filtering steps, we updated our PRIDE submissions to include our Perseus analysis files, which can be viewed and manipulated in Perseus (version 1.6.2.3).

Concerning the nuclear markers for young GCs: Although three proteins may seem to be a low number, we would like to point out that FAMA-stage young GCs are a rare cell type we still have very little information on. At the moment we do not know any other nuclear proteins specific for this cell type. This re-emphasizes the challenges of proteomics with rare cell types and the need for techniques such as PL to identify more proteins specific for (or highly enriched in) the nucleus of this, and other rare, cell types.

3) The extensive data filtering to generate a high confidence sets (especially with regard to FAMA interactors) would usually require validation of some of the new candidates. Have the authors conducted any preliminary validations of the new interactors? Even relatively preliminary data would strengthen their claims for this approach. This could be preliminary transient work e.g. co express FAMA-TurboId and Target:Myc, pull down with streptavidin and conduct a western blot with anti-Myc. This would ideally be transient targets rather than complex partners, as this is the articulated advantage of this method over e.g. AP-MS.

We are in total agreement that validation of candidates is important. We appreciate the reviewers’ suggestion of an experiment that seems straightforward to do without involving stable transformation, but we feel that this experiment, that essentially uses the same proteins but at much higher levels in biologically irrelevant cells, may not be the ideal way to validate the results. In addition, with the publication of (Zhang et al., 2019), some of the concern about the ability of TurboID to detect interactions may have been addressed.

Nonetheless, we did a small-scale test as requested by the reviewer and also used an orthogonal method (Y2H) to test some of the interactions. The data are now presented in Figure 5—figure supplement 5 (subsection “Proximity labeling is superior to AP-MS for identification of candidate interactors of FAMA”, seventh paragraph; subsection “Suitability of TurboID and miniTurbo for application in plants and performance of TurboID in FAMA-complex identification and nuclear proteome analysis”, second paragraph and Materials and methods subsection “Cloning of FAMA interaction partner candidates for Y2H and PL assays in *N. benthamiana*”; subsection “Evaluation of selected FAMA interaction partner candidates by Y2H and PL assays in *N. benthamiana*”). Briefly, in our Y2H, we looked for interaction of FAMA with four candidates recovered from the PL screen: SEUSS (SEU) and LUH (two parts of a co-repressor complex) and BZR1 and BIM1 (two TFs involved in BR signaling), as well as known positive (ICE1) and negative (MUTE) controls. There was clear interaction with BIM1 and SEU and weak interaction with LUH. To gain more insight into the FAMA-SEU-LUH interaction, we co-expressed FAMA-TbID or nuclear TbID with SEU and/or LUH in tobacco and tested for biotinylation of SEU and LUH. To label only close interactors we did not add exogenous biotin. Under these conditions, we could observe biotinylation of ICE1 (pos. control) and SEU, but not of LUH, supporting our hypothesis that SEU is a direct interaction partner of FAMA, while LUH might bind to FAMA via SEU.

4) The access to the raw MS data in PRIDE does not work. Both PRIDE login details resulted in no available data. I would guess data have been updated in the interim and this requires a new username / password to be issued for reviewers. Thus the robustness of the MS data matching employed by the authors could not be evaluated nor examine the quantitative process which is presumably available in a viewable data format.

We are very sorry that the reviewers did not have access to the PRIDE submission. We indeed uploaded additional data before submission, which changed the reviewer account, something we did not realize until receiving the reviewers’ comments. We have double checked that the access information is correct in the newly submitted version, and ask the editor and reviewer to let us know immediately if they have access problems so we can fix them.

We would like to point out that the current version of PRIDE does not allow full data submission for MaxQuant analysis because not all required files are generated by MaxQuant. Our submissions are therefore not in a graphically viewable format in the PRIDE repository. To allow data re-analysis and evaluation we have submitted the RAW files (for re-analysis in any software of choice), the MaxQuant output folder that contains the search summary and search result (to evaluate the data search and do data filtering with a software of choice), as well as our Perseus files containing the analysis path (for graphical visualization of the data and data filtering process in Perseus).

5) Figure 5—figure supplement 2B: Why is replicate 1 is separated from replicate 2 and 3? It seems that separation by replicates is more significant than separation by genotypes. Is this due to how samples were processed? It was not totally clear from the method. Would be nice to have multi scatter plots as well to show correlation between samples.

As we pointed out in the figure legends to Figure 5—figure supplement 2 and Figure 6—figure supplement 1, we have a relatively strong batch effect from sample preparation. Batch effects are not uncommon in mass spec analysis, especially when enrichment steps are involved in sample preparation (which is the case in our experiments), and can sometimes be the strongest separating factor. As this is designed to be a method useful for the community, we chose to be completely transparent about this issue. Due to the number of samples we were unable to process all samples at the same time (a situation others are likely to encounter). We therefore did sample prep in three batches, each of them containing one replicate of each sample type, to avoid introducing a bias from slightly different sample handling in different batches (described in Materials and methods subsection “Affinity purification of biotinylated proteins”).

Concerning the significance of separation, we would like to point out that PCA was done with the full dataset of ‘high confidence’ proteins (present in all replicates of at least one sample type) and the majority of proteins in this dataset (~ 90%) are not significantly enriched in any of the TbID samples. Nevertheless, we already see separation of WT and TbID samples in PC1. To better visualize the high similarity of all samples in the ‘FAMA interactome’ experiment we added a multi scatterplot also displaying the Pearson correlation between sample pairs as suggested by the reviewers (see Figure 5—figure supplement 3). We also added a multi scatterplot for the ‘nuclear proteome’ experiment (Figure 6—figure supplement 2), which shows much stronger differences of individual samples.

6) In my understanding Branon et al. named the new ligases as TurboID(TbID) and miniTurbo(mTb), and therefore authors should use same terms to avoid any confusions.

We have replaced TID with TbID, miniTID with mTb and miniTurboID with miniTurbo throughout the text, in all figures and tables.